# Layerwise complexity-matched learning yields an improved model of cortical area V2

**Nikhil Parthasarathy**  *np1742@nyu.edu*
*Center for Neural Science, New York University*
*Center for Computational Neuroscience, Flatiron Institute*

**Olivier J. Hénaff**  *henaff@google.com*
*Google DeepMind*

**Eero P. Simoncelli**  *eero.simoncelli@nyu.edu*
*Center for Neural Science, New York University*
*Center for Computational Neuroscience, Flatiron Institute*

**Reviewed on OpenReview:** *https://openreview.net/forum?id=lQBsLfAWhj*

## Abstract

Human ability to recognize complex visual patterns arises through transformations performed by successive areas in the ventral visual cortex. Deep neural networks trained end-to-end for object recognition approach human capabilities, and offer the best descriptions to date of neural responses in the late stages of the hierarchy. But these networks provide a poor account of the early stages, compared to traditional hand-engineered models, or models optimized for coding efficiency or prediction. Moreover, the gradient backpropagation used in end-to-end learning is generally considered to be biologically implausible. Here, we overcome both of these limitations by developing a bottom-up self-supervised training methodology that operates independently on successive layers. Specifically, we maximize feature similarity between pairs of locally-deformed natural image patches, while decorrelating features across patches sampled from other images. Crucially, the deformation amplitudes are adjusted proportionally to receptive field sizes in each layer, thus matching the task complexity to the capacity at each stage of processing. In comparison with architecture-matched versions of previous models, we demonstrate that our layerwise complexity-matched learning (LCL) formulation produces a two-stage model (LCL-V2) that is better aligned with selectivity properties and neural activity in primate area V2. We demonstrate that the complexity-matched learning paradigm is responsible for much of the emergence of the improved biological alignment. Finally, when the two-stage model is used as a fixed front-end for a deep network trained to perform object recognition, the resultant model (LCL-V2Net) is significantly better than standard end-to-end self-supervised, supervised, and adversarially-trained models in terms of generalization to out-of-distribution tasks and alignment with human behavior. Our code and pre-trained checkpoints are available at https://github.com/nikparth/LCL-V2.git

## 1 Introduction

Perception and recognition of spatial visual patterns, scenes and objects in primates arises through a cascade of transformations performed in the ventral visual cortex (Ungerleider & Haxby, 1994). The early stages of visual processing (in particular, the retina, lateral geniculate nucleus, and cortical area V1), have been studied for many decades, and hand-crafted models based on linear filters, rectifying nonlinearities, and local gain control provide a reasonable account of their responses properties (Shapley & Victor, 1979; McLean & Palmer, 1989; Adelson & Bergen, 1985; Carandini et al., 1997) Complementary attempts to use bottom-

up normative principles such as sparsity, coding efficiency, or temporal prediction have provided successful accounts of various early visual properties (Atick & Redlich, 1990; Van Hateren & van der Schaaf, 1998; Li, 1996; Olshausen & Field, 1996; Bell & Sejnowski, 1997; Schwartz & Simoncelli, 2001; Karklin & Lewicki, 2009; Wiskott & Sejnowski, 2002; Hoyer & Hyvärinen, 2002; Cadieu & Olshausen, 2012; Karklin & Simoncelli, 2011). But these also have been limited to early stages up to area V1, and have thus far not succeeded in going beyond.

Deep neural networks (DNNs), whose architecture and functionality were inspired by those of the primate visual system (Fukushima, 1980; Douglas et al., 1989; Heeger et al., 1996; Riesenhuber & Poggio, 1999), have offered a new opportunity. When trained with supervised and self-supervised end-to-end (E2E) backpropagation, DNNs have provided the first models that begin to capture response properties of neurons deep in the visual hierarchy (Yamins et al., 2014; Zhuang et al., 2021; Schrimpf et al., 2018; Kubilius et al., 2019). Early results showed that these DNNs are also generally predictive of the overall category-level decisions of primates during object recognition tasks (Ghodrati et al., 2014; Jozwik et al., 2016; Kheradpisheh et al., 2016); however, they have not been predictive of more detailed behavior, as measured by alignment with individual image confusion matrices (Rajalingham et al., 2018). Nevertheless, as the field has rapidly progressed, more recent results demonstrate that scaling end-to-end task-optimization (both in training data and model size) leads to significant improvements in predicting this trial-by-trial human behavior in matched visual tasks (Geirhos et al., 2021; Sucholutsky et al., 2023).

Ironically, despite their historical roots, these same networks have not provided convincing models of early visual areas such as V1 and V2, and do not account for other perceptual capabilities (Berardino et al., 2017; Hénaff et al., 2019; Fel et al., 2022; Subramanian et al., 2023; Bowers et al., 2022; Feather et al., 2023). Figure 1 summarizes these observations for a set of models with a wide variety of architectures and training paradigms, drawn from the BrainScore platform (Schrimpf et al., 2018). The left panel shows that improvements in object recognition performance are strongly correlated ($r = 0.57$) with improvements in accounting for human recognition capabilities. This is encouraging, but perhaps expected, since the recognition databases used for training represent human-assigned labels. The right panel shows that there is also a positive correlation (albeit weaker) between recognition performance and ability to explain responses of IT neurons recorded in macaque monkeys. Again, this is perhaps not surprising, given that object-recognition behavior can be to some extent explained by linear weightings of neurons in inferior temporal (IT) cortex (Majaj et al., 2015). (It is worth noting, however, that for models with very high recognition performance ($>70\%$), the ability to explain IT neurons has gotten progressively worse (Linsley et al., 2023)). Surprisingly, recognition performance is uncorrelated (or even slightly anti-correlated) with the ability to explain responses of visual neurons in cortical areas V1 and V2.

Why do these networks, which offer human-like performance in complex recognition tasks, and which provide a reasonable account of neural responses in deep stages of the visual hierarchy, fail to capture earlier stages? We interpret this as an indication that intermediate DNN layers are insufficiently constrained by end-to-end training on recognition tasks. More specifically, the extremely high model capacity of these networks allows the training procedure to find "shortcuts" that satisfy single end-to-end objectives (both supervised and self-supervised) (Geirhos et al., 2020a; Robinson et al., 2021). As a result, it is common for networks to utilize unreliable feature representations that do not generalize well (Hermann & Lampinen, 2020). This is further evidenced by the fact that standard trained networks can be fooled by 'adversarial examples' (small pixel perturbations that can large shifts in internal classification decision boundaries) (Szegedy et al., 2013; Goodfellow et al., 2014; Tramèr et al., 2017). With this in context, it makes sense that the current best DNN models for V1/V2 seem to be those that are trained to increase robustness to adversarial attacks (Madry et al., 2017). However, the specific solution of adversarial training comes at a significant cost in standard image recognition performance, as well as being computationally expensive. Moreover, this training procedure still propagates gradients via E2E backpropagation, which is generally considered to be biologically implausible.

In this work, we hypothesize that representations throughout a DNN can be constrained in a more biologically-plausible manner through the use of *layerwise* self-supervised learning objectives. We propose a natural method for matching the complexity (or difficulty) of these objective functions with the computational capacity at each stage of processing. When used to train a two-stage model, the resulting network

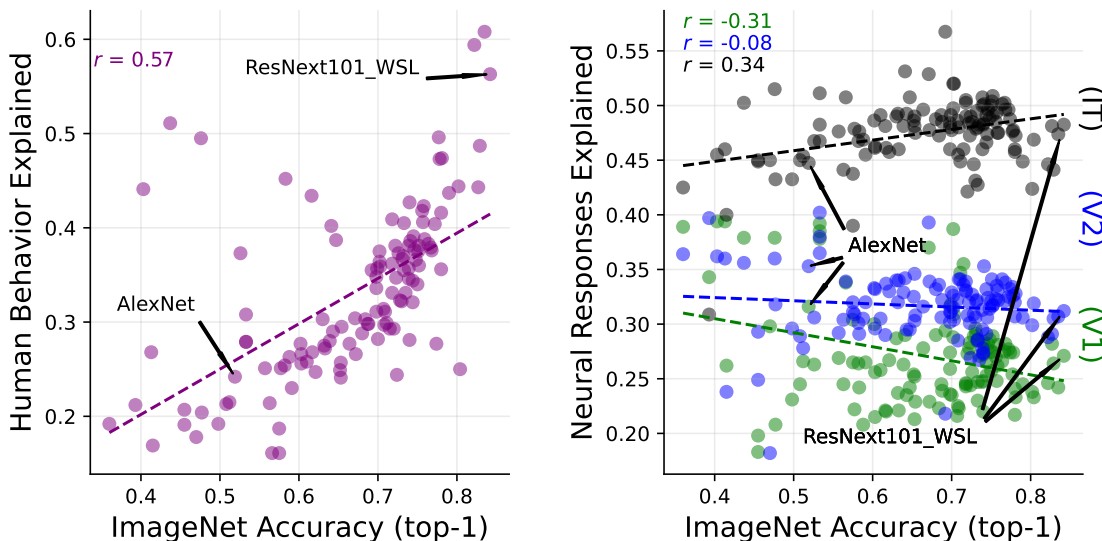

Figure 1: **DNN object recognition performance predicts human recognition behavior, but not primate early visual responses.** Each plotted point corresponds to a DNN model from the BrainScore database (Schrimpf et al., 2018)). Horizontal axis of both panels indicates recognition accuracy (top-1) on the ImageNet dataset (Krizhevsky et al., 2012). **Left:** Comparison to alignment with human visual recognition performance (combination of benchmarks taken from Geirhos et al. (2021) and Rajalingham et al. (2018)). **Right:** Comparison to neural variance explained by regressing the best-fitting DNN layer to neural responses measured in macaque V1 (green), V2 (blue) (Freeman et al., 2013; Ziemba et al., 2016) and IT (black) (Majaj et al., 2015; Sanghavi & DiCarlo, 2021; Sanghavi et al., 2021b;a)

achieves state-of-the-art predictions of neural responses in cortical area V2. Furthermore, when using this learned model as a front-end for supervised training with deeper networks, we show that (in contrast with adversarial training) the increased neural alignment does not come at the cost of object recognition performance, and in fact results in significant improvements in out-of-distribution recognition performance and alignment with human behavior.

## 2 Methods

We first describe the key conceptual underpinnings of our layerwise training method, and then provide the experimental training and evaluation details.

### 2.1 Layerwise complexity-matched learning

Layerwise (more generally, blockwise) methods for DNN training have been previously developed to alleviate the global propagation of gradients required in E2E training training (Hinton et al., 2006; Bengio et al., 2006; Illing et al., 2021; Belilovsky et al., 2019; Siddiqui et al., 2023; Halvagal & Zenke, 2023). Figure 2 illustrates the relationship between the two approaches. Given a set of inputs $x$ and corresponding output labels $z$, the E2E approach optimizes all network parameters $\theta$ to minimize the loss function $L_{E2E}$ via full backpropagation. Successful training of high-capacity networks has generally been achieved with large amounts of training data and complex objectives: (1) supervised data that encourages object-level semantic invariances (Krizhevsky et al., 2012), (2) self-supervised data generated using a combination of spatial and photometric augmentations (Zbontar et al., 2021; Chen et al., 2020b; Grill et al., 2020), or self-supervised masked autoencoding with substantial levels of masking (He et al., 2021). In general, the quality of learned features and the success in recognition depends on the complexity (or difficulty) of the learning problem. For example, if we consider a supervised classification objective, the difficulty of this problem will depend on factors such as the number of classes, complexity of the image content (simple shapes vs. real-world objects),

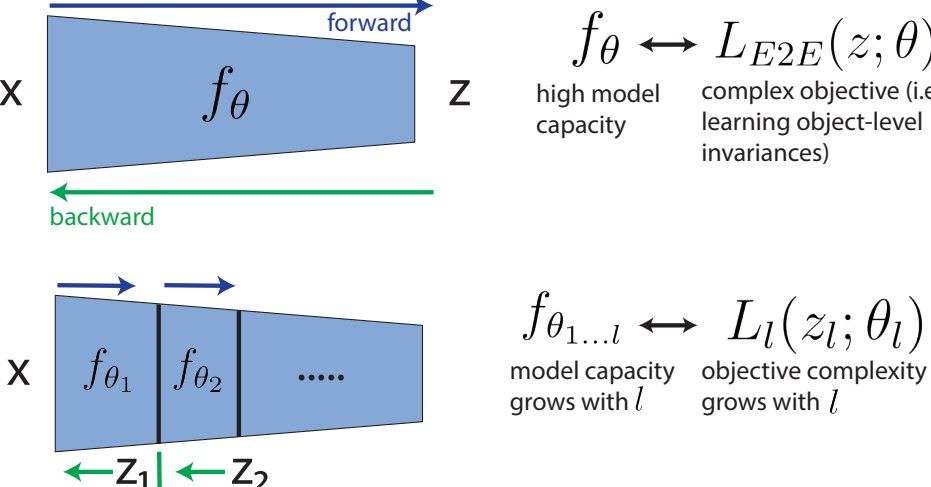

Figure 2: **Layerwise complexity-matched learning**. **Top:** The standard end-to-end (E2E) learning paradigm used with DNNs. The loss function ($L_{E2E}$) operates on the network output and is typically chosen to favor object-level invariances, through supervised training on labelled data or self-supervised training on augmented examples. To solve these E2E objectives, the network $f(\theta)$, must have a high model capacity (sufficiently large number of parameters and non-linearities). **Bottom:** In a layerwise training system, the loss is a function of all intermediate outputs ($z_1$, $z_2$,...). Losses at each layer $L_l$ are used to train each encoder stage $f_{\theta_l}$ independently, with gradients operating only within stages. For effective training, we hypothesize that the loss at each stage, $L_l$, should be matched in complexity to the model capacity defined by the network up to layer $l$.

or the magnitude of the within-class variability (deformations under which each object is seen). Similarly, these training set properties control the complexity of the self-supervised problem (Robinson et al., 2021; Jing et al., 2021). In contrast, in the layerwise approach, the objective is partitioned into sub-objectives that operate separately on the output of each layer, and the optimization thus relies on gradients that propagate within (but not between) layers. The model at a given layer $l$ is composed of all stages up to that layer: $f_{\theta_{1...l}}$. Thus, the computational capacity is low in the early layers (only a few non-linearities and small receptive fields) and increases gradually with each successive layer. To achieve successful training in this scheme, we propose to *match the complexity of the data diversity and objective function with the effective model capacity at a given layer*.

## 2.2 Self-supervised contrastive objective

We construct a layerwise objective based on the "Barlow Twins" self-supervised loss (Zbontar et al., 2021), a feature-contrastive loss that is robust to hyperparameter choices and has recently shown success in blockwise learning (Siddiqui et al., 2023). Briefly, each image $x$ in a batch is transformed into two views, $x^A$ and $x^B$, via randomly selected spatial and photometric deformations. Both views are propagated through an encoder network $f_\theta$ and a projection head $g_\theta$ to produce embeddings $z = g_\theta \circ f_\theta(x)$. We define a cross-correlation matrix over each batch of images and corresponding view embeddings:

$$c_{ij} = \frac{\sum_b z^A_{b,i} z^B_{b,j}}{\sqrt{\sum_b (z^A_{b,i})^2}\sqrt{\sum_b (z^B_{b,j})^2}} \tag{1}$$

where $b$ indexes the batch samples and $i$ and $j$ index the vector components of the projection head embedding. The Barlow Twins objective function is then:

$$L_{BT} = \sum_i (1 - c_{ii})^2 + \lambda \sum_i \sum_{j \neq i} c_{ij}^2 \tag{2}$$

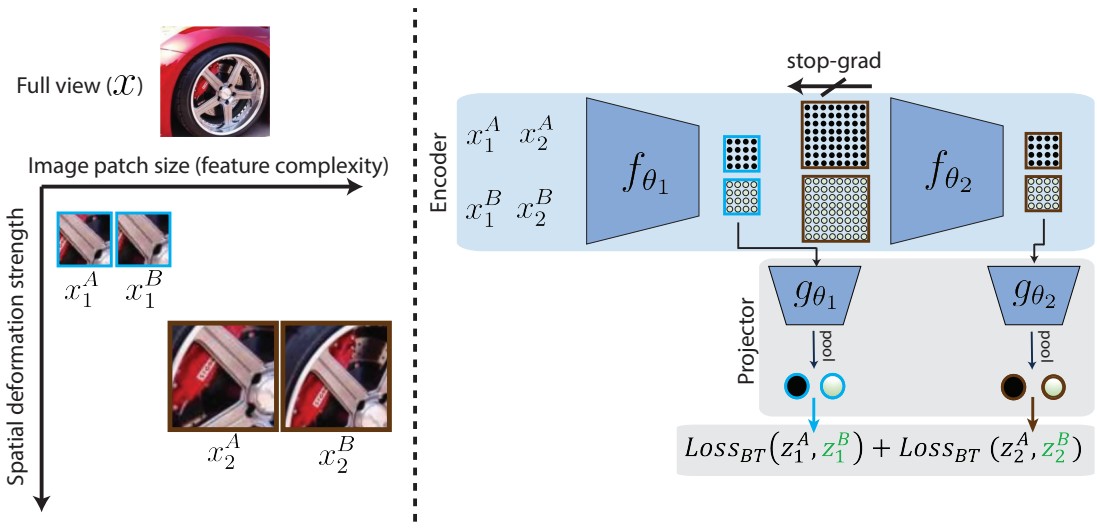

Figure 3: **Layerwise complexity-matched objective. Left:** For each layer, the objective encourages invariance to feature perturbations by comparing the representation of two augmented views of the same image. For layer $l$, the feature complexity of generated image pair $(x_l^A, x_l^B)$ is controlled through choice of patch size, and the magnitude of spatial deformations (translation, dilation). **Right:** The parameters $\theta_1$ of the first layer encoder $f_{\theta_1}$ are updated using the Barlow Twins feature-contrastive loss (Zbontar et al., 2021) operating on the two views of the smallest patch size $(x_1^A, x_1^B)$. This set of views is only propagated to this layer output. The parameters $\theta_2$ of the second layer encoder $f_{\theta_2}$ are updated with the same loss, but using the views that cover a larger spatial region, and include larger spatial deformations.

This loss encourages formation of invariant projection-features (or equivariant encoder features) across the two views (maximizing the diagonal terms of the correlation) while decorrelating or reducing the redundancy of the components of the output vectors (minimizing the off-diagonal terms). This objective can thus be thought of as a "feature-contrastive" method. As noted in Garrido et al. (2022), there is a strong duality between this loss and with sample-contrastive losses (such as SimCLR (Chen et al., 2020b)). Accordingly, we achieve similar results in our framework using sample-contrastive losses (more in Sec. 3.4), but find slight improvements in performance and stability with the Barlow Twins objective.

## 2.3 Training methodology

We apply our **L**ayerwise **C**omplexity-matched **L**earning paradigm (LCL) to a two-stage model, denoted **LCL-V2**. The training methodology is depicted in Fig. 3. The loss function aims to optimize feature invariance across augmented views of an image, while decorrelating features across different images. We control the complexity (difficulty) of each of these learning problems by changing both the size of input images and the strength of augmentation deformations that are used to compute the per-layer loss functions. Fig. 3 (a) depicts this input processing for an example image, considering our two-layer network. Given the full view $x$, a patch is cropped for layer 1 $(x_1^A)$ and layer 2 $(x_2^A)$. We choose an initial patch size for $x_1^A$ and note that the patch size for layer 2 is simply scaled by a factor of 2, roughly matched to the scaling of biological receptive field sizes between areas V1 and V2 (Freeman & Simoncelli, 2011). For the selected patches, we then generate augmentations $(x_2^A, x_2^B)$ using photometric and spatial deformations. For simplicity, we maintain the same photometric deformations and scale the problem complexity by proportionally adjusting the strength of the spatial deformation by a factor of 2 between the two layers. Visually, we see that this procedure results in paired images for layer 1 that have low feature complexity and small translation and scale differences while the images for layer 2 have higher feature complexity and larger deformations.

Given these inputs, Fig. 3(b) shows the procedure for computing the per-layer loss functions. We generate projection embeddings for layer 1 and layer 2 by propagating the corresponding input patches to the

corresponding model blocks:

$$z_1^A = \text{GAP} \circ g_{\theta_1} \circ f_{\theta_1}(x_1^A)$$
$$z_2^A = \text{GAP} \circ g_{\theta_2} \circ f_{\theta_2} \circ f_{\theta_2}(x_2^A)$$

$f_{\theta_l}$ refers to the encoder blocks and $g_{\theta_l}$ corresponds to the projection heads for each layer. We project the embeddings at each spatial location of the feature map independently before applying global average pooling (GAP) [1] to create the final projection embedding. $z_1^B$ and $z_2^B$ are computed analogously from $x_1^B$ and $x_2^B$. The loss is then computed as the sum of losses for each layer: $Loss = L_{BT}(z_1^A, z_1^B) + L_{BT}(z_2^A, z_2^B)$. As in Siddiqui et al. (2023), the loss computation only requires backpropagation within each layer, and gradients from the layer 2 loss do not affect parameters in $f_{\theta_1}$.

In summary, we implement a complexity-matched layerwise learning formulation where the difficulty of the learning problem at layer 2 is scaled in comparison with that at layer 1. The model must learn invariant features across images that have more complex content (larger patch size) that are also more strongly deformed (in scale and translation). This increase in objective complexity accompanies a corresponding increase in model capacity in the second layer (due to growth in receptive field size and number of nonlinearities).

## 2.4 Implementation details

**Architecture.** As in many previously published results (Caron et al., 2018; Gidaris et al., 2018), we chose to use the AlexNet architecture (Krizhevsky et al., 2012) with batch normalization, (Ioffe & Szegedy, 2015). While many recent results make use of more complex architectures (eg, ResNets (He et al., 2016), Vision Transformers (Dosovitskiy et al., 2020) etc.) our method can be more effectively evaluated with a very shallow network, as we can severely restrict model capacity in training these early layers without confounding architectural features such as skip connections, attention blocks etc. Additionally, as mentioned earlier, much of the biological anatomy and computational theories suggest that V2 should be explainable by a single layer of transformation given V1 afferents (El-Shamayleh et al., 2013; Ziemba, 2016). Given that AlexNet models provide relatively strong baseline architectures for V1 (compared with more complex models such as ResNets and vision transformers), we hypothesize that this simple architecture can provide a starting point towards parsimonious and interpretable models of early vision.

For LCL-V2 we train the first two convolutional stages of the AlexNet architecture and utilize a standard multi-layer perceptron (MLP) with a single hidden layer for the projector networks at each layer. The computational capacity is increased between the two stages, with each stage incorporating two non-linearities (ReLU activation and MaxPooling). In addition, capacity is scaled by increasing the number of channels (64 to 192) and receptive field size (via (2x) subsampled pooling). In Sec. 3.5, we additionally evaluate the effectiveness of LCL-V2 as a fixed front-end model (similar to Dapello et al. (2020)). We train the remaining AlexNet layers (with batch normalization) on top of the fixed LCL-V2 front-end and refer to this full network as **LCL-V2Net**. For more specific architecture details, see appendix Sec A.1.

**Data and Optimization.** We train LCL-V2 and its ablations (see Sec. 3.4) on the ImageNet-1k dataset (Krizhevsky et al., 2012). We resize the original images to minimum size 224x224. For layer 1 we centrally crop a 56x56 patch and generated spatially augmented views of size 48x48 via the RandomResizedCrop (RRC) operator with scale = $(0.6, 0.9)$. For layer 2, we central crop a 112x112 patch and generate views of size 96x96 with RRC crops (scale = $(0.3, 0.9)$). As a result, both the final patch size and crop scale range are doubled between layer 1 and layer 2. For each set of patches, we also apply a fixed set of photometric distortions by weakly varying contrast and luminance, and adding random Gaussian noise with variable standard deviation (details in Sec. A.2). Unlike standard E2E self supervised approaches, we do not use the more aggressive augmentations (large color jitter, flipping etc), which seem less perceptually relevant.

We use the Adam optimizer (Kingma & Ba, 2014) without weight decay and $lr = 0.001$. We train the model until the summed validation loss (evaluated on a held-out set of images) does not improve above beyond a fixed threshold. While recent work in self-supervised learning has found benefits from using more complex optimizers and learning rate schedules, we find no significant benefits in our two-layer setting. To train the

---

[1]The additional GAP pooling is only applied during training.

full LCL-V2Net, we fix the pre-trained LCL-V2 as a front-end and use a standard supervised cross-entropy loss to train the subsequent stages. We train for 90 epochs using the SGD optimizer ($lr = 0.1$) with a step-wise learning rate scheduler that reduces the learning rate every 30 epochs.

### 2.5 Experimental setup

**Model comparisons.** Throughout this work we compare to a variety of previous models of three types (see Sec. A.1 for details):

- **E2E (standard)**: End-to-end AlexNet models trained with standard supervised or self-supervised objective functions on the ImageNet-1K dataset: Supervised (Krizhevsky et al., 2012), Barlow Twins (Zbontar et al., 2021), and VOneNet (fixed-V1 stage and supervised learning for downstream stages) (Dapello et al., 2020).

- **E2E (robust)**: End-to-end AlexNet models trained with state-of-the-art robustification methods specifically to maintain robustness to adversarial pixel perturbations: standard adversarial training (L2-AT ($\epsilon = 3.0$)) (Madry et al., 2017), adversarial noise training with a parameterized noise distribution (ANT) (Rusak et al., 2020).

- **Layerwise training**: AlexNet model trained with the Barlow Twins objective using thestandard image augmentation scheme applied layerwise (Siddiqui et al., 2023), Latent-predictive-learning (LPL)(Halvagal & Zenke, 2023).

- **Hand-crafted**: Steerable pyramid layer (Simoncelli & Freeman, 1995) (with simple and complex cell nonlinearities), followed by a layer of spatial $L_2$ (energy) pooling.

**Neural alignment evaluations.** We compare all models quantitatively in their ability to predict aspects of V2 neurons from the dataset used in BrainScore (Schrimpf et al., 2018). This dataset, described in Freeman et al. (2013); Ziemba et al. (2016), provides electrophysiological recordings of 103 V2 neurons responding to texture images synthesized with the Portilla-Simoncelli texture model (Portilla & Simoncelli, 2000). The data include responses to 15 texture samples, in addition to 15 samples of spectrally-matched noise images, for 15 different texture families (a total of 450 images). To measure model predictivity of this neural data, we fit models with the data splits and implementation of the partial least squares (PLS) regression method proposed in Schrimpf et al. (2018), and then compute explained-variance scores for each fitted model. Details are provided in Sec. A.3.

To better understand the ability of models to capture selectivities of V2 neurons, we provide additional evaluations (Sec. 3.2) that use the texture modulation ratio statistic introduced in Freeman et al. (2013). Specifically, we define: $R_{mod_{n,i}} = \frac{tex_{n,i} - noise_{n,i}}{tex_{n,i} + noise_{n,i}}$, where $tex_{n,i}$ is the response of neuron $n$ (averaged across 15 image samples) to texture family $i$ and $noise_{n,i}$ is the corresponding response to the spectrally-matched noise for family $i$.

**LCL-V2Net recognition and human behavior evaluations.** We primarily use the out-of-distribution (OOD) generalization benchmark of Geirhos et al. (2021) to test the performance of LCL-V2Net. This dataset consists of 17 OOD classification tasks based on adding various kinds of noise, distortions, and shape-biasing transformations to ImageNet images. We evaluate both OOD accuracy and consistency with human behavior. For more information on the benchmark, specific list of distortions, and evaluation metrics see Sec. A.4.

We additionally report performance on the original ImageNet-1K (Krizhevsky et al., 2012) validation set as well as more recent large-scale validation sets (ImageNet-R (Hendrycks et al., 2021) and ImageNet-vid-robust (Shankar et al., 2021)) for testing generalization.

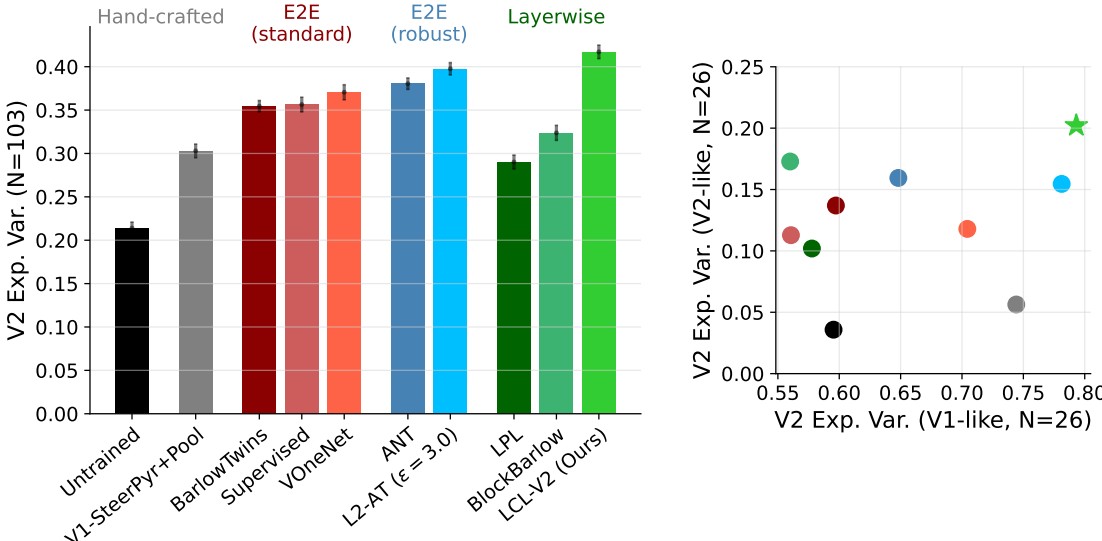

Figure 4: **The LCL-V2 model outperforms other models in accounting for V2 responses. Left:** Median explained variance of models fitted with PLS regression to 103 primate V2 neural responses. For models with more than two layers, all layers are evaluated and the performance of the best layer is provided. Standard deviations over 10-fold cross-validated regression are indicated on each bar. **Right:** Comparison of median explained variance for "V1-like" and "V2-like" V2 cells. These categories correspond to the top and bottom quartiles (N=26) of V2 cells sorted by how well they are fit by a canonical hand-constructed V1 model (V1-SteerPyr+Pool). The minimum explained variance of the V1 model over the set of "V1-like" neurons is 57 %. The maximum explained variance of the V1 model over the set of "V2-like" neurons is 4 %. The LCL-V2 and L2-AT models significantly outperform all other models on the V1-like subset, even surpassing the baseline V1 model. The LCL-V2 model also significantly outperforms the L2-AT model on the V2-like subset.

## 3 Results

### 3.1 Population fits to neural data

**Overall V2 predictivity.** The first panel of Fig. 4 shows the overall BrainScore explained variance of the models outlined in Sec. 2.5. For all models, the best layer was chosen by evaluating predictions on a validation set prior to fitting the final PLS regression on the held-out test set. We see that LCL-V2 outperforms all architecture-matched models, including the L2-AT trained network. In fact, although we only show architecture-matched results here, our model provides the best account of the V2 data across all architectures currently[2] on the BrainScore leaderboard (Schrimpf et al., 2018; 2020). We provide the detailed numbers in Table A.4. Interestingly, previous layerwise training methods ('Barlow (layerwise)' (Siddiqui et al., 2023) and 'LPL' (Halvagal & Zenke, 2023)) exhibit significantly worse performance than the standard end-to-end training. This suggests that the benefits of our method specifically arise from complexity-matching, which we quantify further in Sec. 3.4.

**Partitioning V2 with a V1-baseline model.** The V1-SteerPyr+Pool model provides a baseline measure of how well V2 neurons can be predicted simply by combining rectified and $L_2$-energy pooled oriented filter responses (as are commonly used to account for V1 responses). Nearly 30 % of the variance across all 103 V2 neurons can be explained given this model, suggesting that there are a number of V2 neurons that are selective for orientation and spatial frequency selectivity. In fact, this aligns with prior studies that have found subsets of V2 neurons with tuning similar to V1 neurons (but with larger spatial receptive fields) (Foster et al., 1985; Levitt et al., 1994; Willmore et al., 2010; Lennie, 1998).

---

[2]as of publication date: 06/2024

Given this baseline model, we partition the V2 neural datasets into neurons that are 'V1-like' (top quartile, in terms of how well they are explained by the V1-SteerPyr+Pool model) and those that are 'V2-like' (bottom quartile). In the right panel of Fig. 4, we compare the median performance of each of the models on these two subsets. We find that all other non-adversarially trained models (both layerwise and end-to-end) are significantly worse at predicting the 'V1-like' subset than the baseline V1 model. Surprisingly, both LCL-V2 (ours) and L2-AT models outperform the V1 baseline on this subset, suggesting that although these neurons are most-likely orientation and spatial frequency tuned, they also have some selectivity that is not captured by the simple V1 model. On the 'V2-like' subset, the performance of all models is significantly worse; however, there is now an even larger gap (approx 5%) between LCL-V2 and the L2-AT model. Thus, the adversarial training achieves better predictions of area V2, primarily by better explaining the neurons that have 'V1-like' properties. LCL-V2 maintains this improvement, but also provides better fits to neurons whose complex feature selectivity is not well described by the baseline V1 model. In the following section, we examine whether the models exhibit known feature selectivities found in the V2 data.

**Per-neuron V2 predictivity.** To explore whether the above summary statistics are representative of the predictions of a given model at the per-neuron level, we also computed the predictions of our model for each of the 103 V2 neurons. When compared against predictions of the L2-AT and Supervised AlexNet models, we find that there is in fact a wide distribution of explained-variances across neurons but that our model outperforms the two other models on a majority of neurons (see Fig. A.2 for more details). We believe it could be quite valuable in future studies to further analyze this data to understand what properties of individual neurons determine the variability in predictions of a given model.

### 3.2 Model comparisons via texture modulation

Cortical area V2 receives most of its input from V1. A fundamental property of V2 neural responses that is not present in V1 responses is that of *texture modulation* (Freeman et al., 2013; Ziemba et al., 2016), in which responses to homogeneous visual texture images are enhanced relative to responses to spectrally-matched noise. As described in Sec. 2.5, we compute a texture modulation index $R_{mod_{n,i}}$ for each of the 103 neurons, for each of the 15 texture families. We computed the same modulation index for each neuron in the selected V2-layer from each computational model. In Fig. 5 we compare LCL-V2 against the top two other *fully learned* models in terms of overall V2 explained variance (L2-AT and standard ImageNet1K-Supervised). We exclude the VOneNet model here as it uses a fixed front-end with a different architecture.

We first compare the three models in terms of their ability to capture the full distribution of texture modulation ratios in the V2 dataset (Fig. 5(a)). We compute a modulation ratio for each neuron by averaging over texture families: $R_{mod_n} = \frac{1}{T} \sum_{i=1}^{T} R_{mod_{n,i}}$. We use a quantile-quantile (Q-Q) plot to compare the quantiles of the distribution of these values to those arising from the modulation ratios of each fitted model neuron. It is visually clear that while none of the models perfectly match the V2 neural distribution, LCL-V2 is significantly closer than the other two.

Next, we compute texture modulation ratios for each texture family by averaging over neurons $R_{mod_i} = \frac{1}{N} \sum_{n=1}^{N} R_{mod_{n,i}}$. Because different texture classes have different types of feature content, they stimulate V2 neurons differently, relative to their spectrally-matched counterparts. We compare the rank-ordering of modulations ratios over the texture families for the model and real neurons and scatter-plot the ranks against each other (Fig. 5(b)). The texture family ranks of the LCL-V2 model are well-aligned with those of the actual V2 neurons, whereas both L2-AT and Supervised models yield ranks with many more outliers. This is quantified by the Spearman rank correlation for LCL-V2 ($\rho = 0.90$), which is significantly higher than that of the other two models ($\rho = 0.56$, $\rho = 0.58$). It is worth noting that Laskar et al. (2020) find that this rank correlation can be improved for models by incorporating a subset selection procedure to restrict the specific model neurons used in the comparison.

In summary, although the L2-Robust and Supervised models provide competitive predictivity of the V2 neural responses (Fig. 4), the LCL-V2 model provides a better account of the texture selectivity properties of these neurons.

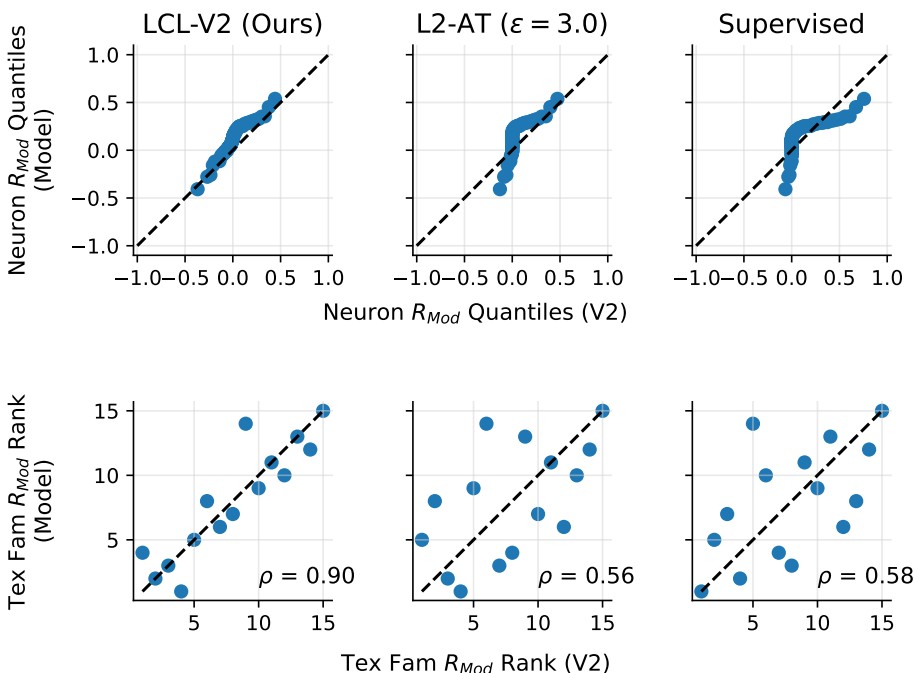

Figure 5: **The LCL-V2 model outperforms other models in capturing texture modulation properties of V2 neurons.** We compare the top 3 (in terms of overall V2 predictivity) fully-learned models: LCL-V2 (Ours), L2-AT, and Supervised. **Top:** Quantile-quantile (Q-Q) comparison of the distribution of texture modulation index values ($R_{mod}$, averaged over texture families) for real and model neurons. LCL-V2 shows better alignment with the physiological distribution (closer to the identity line (dashed)) than the other two models. **Bottom:** Comparison of texture modulation indices for each of 15 texture families (averaged over neurons). The modulation indices for model and real neurons are ranked (1 = lowest modulation family, 15 = highest modulation family), and plotted against each other. Our model provides better alignment with the V2 data, achieving a Spearman rank correlation of $\rho = 0.9$. P-values were computed to test significance of the *difference between the Spearman correlations* using the methodology described in Sec. A.3.3. We find the difference to be significant vs. both models (L2-AT, Supervised) with (p = 0.040, p=0.047) respectively.

### 3.3 V1 layer analysis

To demonstrate the generality of our LCL approach in learning feature hierarchies, we evaluate the first-stage (LCL-V1) in terms of alignment with V1 responses and selectivities. In Fig. 6, we find that the LCL-V1 model outperforms all non-adversarially trained models in terms of V1 explained variance (approx 2-4 % improvement on average), and is on par with both adversarially-trained models (ANT and L2-AT). Furthermore, when visualizing and characterizing the learned receptive fields, we find reasonable qualitative similarity with receptive field properties extracted from the V1 data in (Ringach, 2002) (for details, see Sec. A.5). Note, however, that the hand-crafted models (VOneNet and V1-SteerPyr) still provide better accounts for the V1 responses than all learned models. We suspect this is largely due to the limited receptive field sizes of the learned AlexNet-based models, all of which use a single convolutional layer with 11x11 kernels.

### 3.4 Ablations

We now examine the effect of ablations of our architectural and training choices on physiological alignment.

**Complexity-mismatch.** Fig. 4 shows a significant improvement in V2 predictivity over all non-adversarially trained models and in particular improves dramatically over the application of the layerwise training approach outlined in Siddiqui et al. (2023). Since that model was also trained with the Barlow Twins

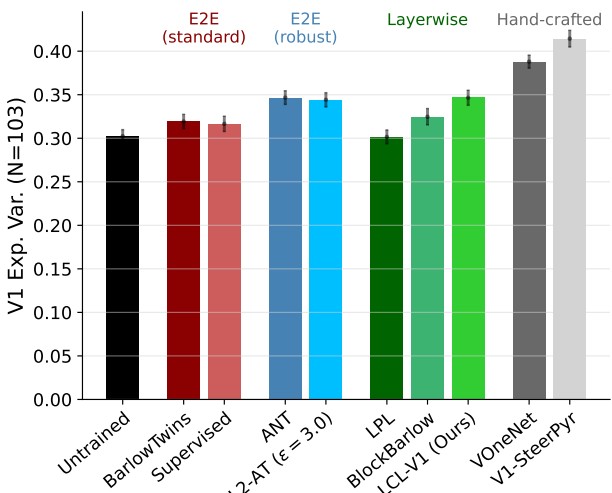

Figure 6: **LCL-V1 outperforms learned models in V1 predictivity and approaches the performance of hand-tailored V1 models.** Analogous to the V2 comparisons (Fig. 4), we also evaluate the best model layers for explaining the V1 neural responses from the same dataset. The highest explained variance is obtained by the hand-designed V1-SteerPyr and VOneNet models. The LCL-V1 model performs similarly to the adversarially robust models, and outperforms all other trained models. Standard deviations over 10-fold cross-validated regression are indicated for each model.

objective, the primary difference with our model is in the complexity-matching of our objective. Specifically, while they use at each layer a set of augmentations generally used to train large object recognition networks, we approximately complexity-match the objective with capacity at each stage of our model. To better understand the quantitative impact of this, we evaluate the neural predictivity of our learned model when introducing complexity-mismatch via changes in the patch size (feature complexity) or random crop scale (spatial deformation strength). Fig. 7 shows that the optimal performance (in terms of neural predictivity) is achieved only when the relative complexity is matched between the two layers. Specifically, we first see that there is an optimal patch size (48px) and deformation strength (s=1) which produces the most aligned V1 layer. This follows from the hypothesis that the the layer 1 views should contain simple edge-like content with small scale deformations. More importantly, once the optimal parameters for the V1 layer training are fixed, we find that both patch size and spatial deformation strength must be scaled accordingly to achieve the optimal V2 model (highlighted in green). This again justifies the initial choice of these scaling parameters based on the natural approximate doubling of receptive field size between V1 and V2.

**Contrastive vs non-contrastive losses.** While numerous self-supervised learning objectives have been proposed over the years, they generally can be classified as contrastive or non-contrastive. While the Barlow Twins loss is 'feature-contrastive', there have been studies demonstrating a duality with 'sample-contrastive' approaches (Garrido et al., 2022; Balestriero & LeCun, 2022). Non-contrastive losses; however, resort to very different mechanisms for avoiding collapsed solutions, with most using some form of 'stop-gradient' based method with asymmetric encoder networks (Chen et al., 2020b; Grill et al., 2020). In Fig. 7(c) we show that some form of a contrastive term (either Barlow Twins or SimCLR) is necessary for achieving optimal neural alignment (especially in the second stage). For example, using SimSiam (Chen et al., 2020b) as a representative non-contrastive loss, greatly hurts the V2 predictivity from the second stage. Therefore, in this context, it seems that in addition to the problem of learning invariances, it is also necessary to include feature decorrelation or sample discrimination as part of the final objective.

### 3.5 OOD object recognition and human-alignment

In the spirit of Dapello et al. (2020), we hypothesized that a more biologically-aligned model of early visual areas (specifically area V2) may provide additional benefits for both recognition performance and alignment with human behavior on visual tasks. Note that while our ultimate goal is to train a full model of the

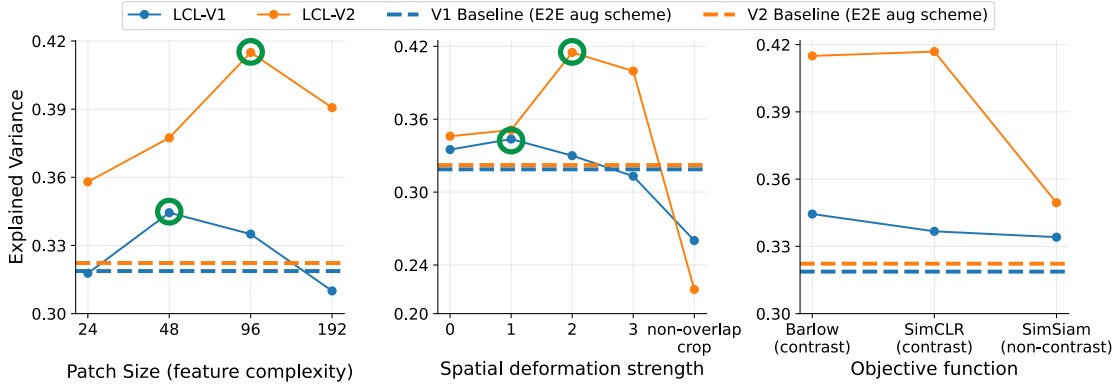

Figure 7: **Substituting complexity-mismatched or non-contrastive objectives decreases neural alignment.** For complexity ablations (left and middle panels), we vary the patch size or spatial deformation strength for the V1 layer (LCL-V1). We then hold these parameters at the optimal values for the V1 layer fixed and again vary the parameters used for training the V2 layer. **Left:** Indicated by the green circle, we see that there is an optimal patch size (feature complexity) for best V1 prediction at 48px and the optimal patch size for V2 is then scaled accordingly (factor of 2 larger). **Middle:** We see that there is also an optimal spatial deformation strength for each layer that is also scaled by a factor of 2 (in minimum crop scale) between each layer. Spatial deformation strength 0 refers to no spatial deformation. Deformations (1-3) refer to the minimum random resized crop scale of (0.6, 0.3, 0.08). 'Non-overlap crop' refers to only using non-overlapping crops. **Right:** We ablate the loss function used to train each layer. We find that performance is very similar with SimCLR, but significantly worse when using a non-contrastive method like SimSiam. Baseline comparisons (dashed lines) indicate performance of the layerwise Barlow method proposed in Siddiqui et al. (2023), which uses the end-to-end training augmentation scheme (details in Sec. A.2) from Zbontar et al. (2021) for each layer.

ventral stream in a layerwise fashion, currently, this would require collecting significantly larger images to capture increasing levels of feature complexity. We therefore opt for a simpler experiment and learn a cascade of additional AlexNet stages appended to the *fixed* LCL-V2 front-end model via the standard ImageNet-1k supervised object recognition task. We then evaluate the full network on the benchmark proposed by Geirhos et al. (2021) which tests both out-of-distribution (OOD) generalization and prediction of human behavior on this task. Again, we refer to this trained recognition network as LCL-V2Net.

**Object recognition accuracy.** We first evaluate the accuracy of our trained network on the ImageNet-1K (Krizhevsky et al., 2012) validation set as well as OOD image set proposed in Geirhos et al. (2021). Fig. 8 (Left) shows that LCL-V2Net significantly outperforms all architecturally-matched (AlexNet-based) models in OOD accuracy by a large margin (4-10 % improvement - see Table A.5). This is particularly striking because the other robust models (VOneNet and L2-AT) do not exhibit a similar improvement, suggesting a potential link between the improved V2 predictions of the LCL-V2 front-end and the generalization of recognition over shifts in the data distribution.

**Human behavioral consistency.** In addition to absolute recognition performance, we also evaluate the ability of the same models to capture human behavioral performance on the same recognition task. The right panel of Fig. 8 shows that LCL-V2Net has significantly better error consistency with the per-trial human recognition decisions. Compared with standard supervised training (consistency=0.165), the only models to show significant improvement are those trained for adversarial robustness (ANT and L2-AT). These models each achieve a consistency of 0.176 (a 6.6 % relative improvement). Without the computational overhead of adversarial training procedures, LCL-V2Net achieves a consistency of 0.211 (a 28 % relative improvement).

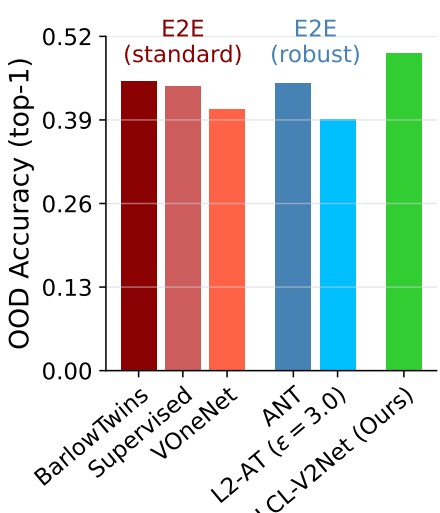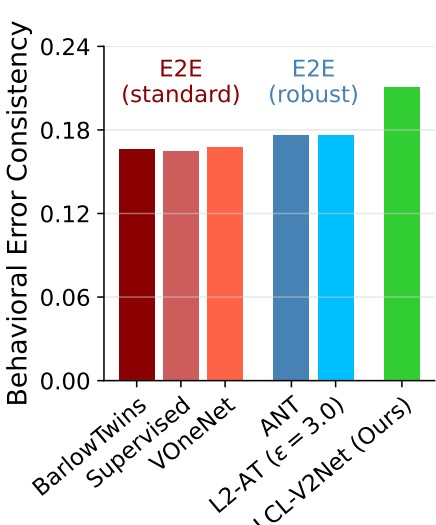

Figure 8: **LCL-V2Net improves OOD generalization and human behavior error consistency. Left:** Supervised training of the later layers of an AlexNet modelon top of LCL-V2 leads to significantly increased OOD accuracy compared to all other architecture-matched models including those trained with standard self-supervised and supervised objectives as well as those trained for robustness. **Right:** LCL-V2Net also shows significantly increased human behavioral alignment, as measured by error consistency on the OOD recognition task. For reference, on this task, humans achieve OOD accuracy of 0.73 and human-human error consistency of 0.43.

## 4 Discussion

We have developed a novel normative theory for learning early visual representations without end-to-end backpropagation or label supervision. We hypothesize that the reason why state-of-the-art DNNs have failed to predict responses of neurons in early visual areas is because they are insufficiently constrained. We then proposed a solution that imposes these constraints through layerwise complexity-matched learning (LCL) that leverages a canonical self-supervised objective at each layer. When applied in a two-stage architecture (LCL-V2), we showed that our trained model is more effective in predicting V1 neural responses in the first layer than other architecturally-matched models, and achieves state-of-the-art quantitative predictions of V2 neural responses. Furthermore, we demonstrated that the LCL-V2 model can be used as a fixed front-end to train a supervised object recognition network (LCL-V2Net) that is significantly more robust to distribution shifts and aligned with human recognition behavior.

As described in the introduction, there is a substantial literature on using normative principles such as sparsity, redundancy reduction, coding efficiency, or temporal prediction to explain early visual properties. These approaches have been mostly limited to early stages (up to and including cortical area V1), and those that have shown some qualitative success in reproducing V2-like selectivities, have again not scaled well beyond small image patches (Willmore et al., 2010; Hosoya & Hyvärinen, 2015; Rowekamp & Sharpee, 2017; Bányai et al., 2019) or have been restricted to texture images (Parthasarathy & Simoncelli, 2020). These are significant limitations, given recent work showing the importance of training on diverse natural image datasets for achieving strong biological alignment (Conwell et al., 2022). We show that a normative approach driven by redundancy reduction (through the Barlow Twins objective) can overcome some of these limitations under our LCL framework. In Sec. 3.4, we also show similar results using the SimCLR objective, but we would like to do a more comprehensive analysis of how more recent promising self-supervised objectives (Yerxa et al., 2023; Assran et al., 2023) perform. More generally, it will be interesting to see whether our

results rely on a form of redundancy reduction or if other principles like sparsity or temporal prediction can be leveraged in a similar manner.

End-to-end trained networks (both supervised and self-supervised) have provided strong accounts of neural responses in late stages of primate cortex as well as recognition behavior (Schrimpf et al., 2018; Geirhos et al., 2021; Zhuang et al., 2021). While unconstrained task-optimization of these models has been the standard for many years, recent efforts demonstrate that constraints on model capacity can lead to better alignment with aspects of biological representation. For example, Nayebi et al. (2023) show that self-supervised (contrastive) E2E training of shallow-networks account well for neurons in mouse visual cortex (due to the limited capacity nature of mouse visual cortex). In primate visual cortex, Margalit et al. (2023) have shown that self-E2E objectives coupled with a layerwise spatial-smoothness regularizer over neural responses produce topographically-aligned models of both primary visual cortex and IT cortex. In contrast with these studies, we hypothesize that E2E objectives do not sufficiently constrain intermediate representations, and that such constraints are better imposed locally, via per-layer objective functions that do not propagate gradients between layers.

In the machine learning literature, there are several examples of layerwise training of DNNs (Belilovsky et al., 2019; Löwe et al., 2019; Xiong et al., 2020; Siddiqui et al., 2023). These efforts have been primarily focused on demonstrating that layerwise objectives can approximate the performance of corresponding end-to-end backpropagation when evaluating on downstream visual tasks. A few studies (Illing et al., 2021; Halvagal & Zenke, 2023) have emphasized the biological plausibility of layerwise learning from a theoretical perspective, but were not scaled to large-scale training datasets. More importantly, previous studies have not assessed whether these biologically-plausible layerwise learning objectives result in more biologically-aligned networks. Here, we've shown that previous layerwise learning approaches (layerwise Barlow (Siddiqui et al., 2023) and LPL (Halvagal & Zenke, 2023)), do not offer the same benefits as our framework. We further demonstrate that a canonical feature-contrastive objective only leads to improved biological alignment *when the objective is complexity-matched with the corresponding computational capacity of the model stage* (See Sec. 3.4).

Dapello et al. (2020) have shown that a biologically-inspired V1 stage greatly improves the adversarial robustness of trained networks. But, a closer analysis of their results (along with the evaluations presented here) demonstrate that a V1-like front end does not in fact provide significantly better robustness to image distribution shifts or alignment with human object recognition behavior. On the other hand, there have been a number of published networks that demonstrate greatly improved behavioral error consistency (Sec. 3.5) at the expense of worse alignment with biological neural responses. These include model scaling (Dehghani et al., 2023), training with natural video datasets, (Parthasarathy et al., 2023), and use of alternative training paradigms (Jaini et al., 2023; Xie et al., 2021b; Radford et al., 2021). Our work provides a step towards resolving this discrepancy, by providing an improved model of early visual areas (specifically area V2) that is accompanied by a corresponding improvement in model generalization and behavioral alignment.

## 5 Limitations and Future Work

We briefly describe some of the limitations of this work and opportunities for future work. First, while we have explored the benefits of layerwise training in a two-stage model, there is opportunity to explore extensions to learning of stages deeper in the visual hierarchy. In order to appropriately scale both the feature complexity (image field-of-view) and spatial deformation strength effectively in more layers, we will need to leverage either larger, scene-level images (Xie et al., 2021a) or natural video datasets (Gordon et al., 2020; Parthasarathy et al., 2023). For many years, improvements in task performance of deep networks was correlated with improved predictions of neurons in late visual areas, but the most recent task-optimized networks have shown a degradation of neural predictivity (Linsley et al., 2023). The extension of our method to deeper layers may help to address these inconsistencies.

Second, the examples and comparisons in this article focused on a single architecture (AlexNet), but we believe the complexity-matching property is of broader applicability. Extending it, however, will require development of more quantitative measures of 1) image content complexity that can be used in place of the current 'patch-size' proxy, and 2) computational capacity of a neural network stage, depending on the specific computations (e.g., number and size of filter kernels, choice of nonlinearity, etc). This is especially important

for extending to recent alternative architectures such as residual (He et al., 2016) or transformer (Dosovitskiy et al., 2020) networks. These contain additional computational elements such as "skip connections" and spatially-global computations, making it difficult to appropriately complexity-match an objective with a given layer in these architectures.

Finally, from a neuroscience perspective, we see a number of opportunities for enhancing and extending the current framework. On the theoretical side, although our layerwise learning is arguably more biologically-plausible than standard supervised, self-supervised, or adversarial training, it still relies on implausible within-layer dependencies. We hope to leverage recently developed methods (e.g., Illing et al. (2021)) to bridge this gap in biological plausibility. Experimentally, the current results are also limited to a few evaluations on a dataset of about 100 neurons and their responses to naturalistic texture and spectrally matched noise images (Freeman et al., 2013; Ziemba et al., 2016). While this dataset is informative, it will be important to compare the LCL-V2 model to V2 responses on a more diverse set of stimuli. Perhaps more exciting is the possibility that elucidation of the structure and response selectivities of the learned LCL-V2 model will reveal new organizing principles for understanding the mysteries of primate visual area V2 and beyond.

### Acknowledgments

We thank Corey Ziemba and the Movshon lab at NYU for providing access to the raw electrophysiology dataset used in this study. We thank Martin Schrimpf for help with and access to the BrainScore implementation and data splits used in the official benchmark. We also thank Pierre-Étienne Fiquet, Manu Raghavan and Thomas Yerxa for helpful discussions during the development of the work. This work was partially supported by NIH grant R01 EY022428 to EPS.

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

# A   Appendix

## A.1   Architecture details

We summarize the details of all architectures used in both the biological-alignment evaluations and the human behavior evaluations.

### A.1.1   Architectures

For the LCL-V2 two-stage model, and or for all of the ablation studies in Sec. 3.4, we use the following architecture:

| | |
|---|---|
| Layer 1 | **Conv2d**(in_channels=3, out_channels=64, ks=11, stride=4, padding=5, padding_mode='reflect')
**BatchNorm2d**(64)
**ReLU**()
**BlurMaxPool2d**(ks=3, stride=2, padding=1) |
| Layer 2 | **Conv2d**(in_channels=64, out_channels=192, ks=5, stride=1, padding=2, padding_mode='reflect')
**BatchNorm2d**(192)
**ReLU**()
**BlurMaxPool2d**(ks=3, stride=2, padding=1) |

Table A.1: **LCL-V2 Two-layer Architecture.** We use the same channel dimensions and non-linearities take from the first two layers of the AlexNet architecture (along with BatchNorm layers). In addition, because we train our model with small image patches, aliasing artifacts impact model responses more than with large images. As a result, we replace standard MaxPooling with anti-aliasing blurring followed by max-pooling (as done in Zhang (2019))

.

As shown in Fig. 3, we use projector networks $g(\theta_1), g(\theta_2)$ during training to encourage learning of 'equivariant' representations:

| | |
|---|---|
| $g(\theta_1)$ | **Linear**(in_channels=64, out_channels=64)
**BatchNorm2d**(64)
**ReLU**()
**Linear**(in_channels=64, out_channels=2048) |
| $g(\theta_2)$ | **Linear**(in_channels=192, out_channels=192)
**BatchNorm2d**(192)
**ReLU**()
**Linear**(in_channels=192, out_channels=2048) |

Table A.2: **LCL-V2 projector network architectures.** Projector networks used during self-supervised pre-training for each layer are MLP networks with single hidden layers. Following Siddiqui et al. (2023), we use output dimensionalities of 2048 for each projector.

For the full LCL-V2Net architecture of Sec. 3.5, we fix the LCL-V2 block and train the following subsequent stages:

| | |
|---|---|
| Downstream features | **Conv2d**(192, 384, ks=3, padding=1) |
| | **BatchNorm2d**(384) |
| | **ReLU**() |
| | **Conv2d**(384, 256, ks=3, padding=1) |
| | **BatchNorm2d**(256) |
| | **ReLU**() |
| | **Conv2d**(256, 256, ks=3, padding=1) |
| | **BatchNorm2d**(256) |
| | **ReLU**() |
| | **MaxPool2d**(ks=3, stride=2) |
| Classifier | **AdaptiveAvgPool2d**(1,1) |
| | **Dropout**(p=0.5) |
| | **Linear**(D, 4096) |
| | **BatchNorm2d**(4096) |
| | **ReLU**() |
| | **Dropout**(p=0.5) |
| | **Linear**(4096, 4096) |
| | **BatchNorm2d**(4096) |
| | **ReLU**() |
| | **Linear**(4096, 1000) |

Table A.3: **LCL-V2Net Downstream Architecture.** On top of the LCL-V2 stage, we train the subsequent stages indicated here based on the AlexNet architecture. We include BatchNorm layers as we find this speeds up convergence.)

.

### A.1.2 V1-SteerPyr+Pool baseline (Hand-crafted)

For the neural response prediction evaluations, we implement a baseline model using the Steerable Pyramid (Simoncelli & Freeman, 1995). We use a 5 scale, 4 orientation complex pyramid (based on 3rd-order oriented derivative filters, and their Hilbert Transforms) as implemented in the Plenoptic package (Duong et al., 2023). We rectify (ReLU) both the real and imaginary channels to generate a total of 40 'simple cells'. We additionally create 20 'complex-cell' channels by computing the modulus of each complex-valued filter response: $r_{complex} = \sqrt{r_{real}^2 + r_{imag}^2}$. For the V1-baseline model, we subsample the output spatial feature map by a factor of 4 to reduce the total number of responses. For the V2-baseline model, we use the response of the V1-stage after applying $L_2$ spatial energy pooling in a channel-independent way with a 3x3 kernel and additional subsampling by a factor 2. Given a 200x200 grayscale input image, the V1-layer response vector therefore has shape (60, 50, 50) while the V2-layer response vector has shape (60, 25, 25).

### A.1.3 VOneNet (Hand-crafted + E2E)

We use the VOneNet-AlexNet network developed in Dapello et al. (2020). We note that this network does not use BatchNorm layers; however, due to inability to re-train this model effectively we use the published pre-trained version. Briefly, this network consist of a VOneBlock front-end model that contains 256 channels (128 simple cell, 128 complex cell) created from a Gabor filter bank basis set with parameters sampled from distributions of recorded macaque V1 cells. This front-end is followed by the full standard AlexNet network architecture from the Pytorch library (Paszke et al., 2019).

### A.1.4 BarlowTwins (layerwise)

We use the same architecture as our LCL-V2 block defined above. However, we utilize the E2E augmentation training parameters defined in Zbontar et al. (2021) and Siddiqui et al. (2023) (see Sec. A.2 for details).

We use this model for comparison in both neural response prediction (Sec. 3.1) and as a baseline for model ablations (Sec. 3.4).

### A.1.5 LPL (layerwise)

We use the training code (`https://github.com/fmi-basel/latent-predictive-learning`) and methods provided in Halvagal & Zenke (2023). While we attempted to re-train the LPL method on the ImageNet-1k dataset, we found that this network would not converge. As a result, we use the AlexNet model trained on the STL-10 dataset for 800 epochs (as done in the original work).

We only use this model for comparison in the neural response prediction experiment (Sec. 3.1) to provide an additional layerwise learning baseline.

### A.1.6 BarlowTwins (E2E)

Due to the fact that there is not an available pre-trained AlexNet-based version of the Barlow Twins E2E training method from Zbontar et al. (2021), we pre-train our own version based on the standard AlexNet architecture (with BatchNorm layers after each convolution/linear layer). We pool the final 'feature' layer such that each image is represented by a single 256-d responses vector. As implemented in Zbontar et al. (2021), before the loss computation, this vector is propagated through a standard MLP projector network with 2 hidden layers. We varied different projector sizes and found the best projector setting to be one with dimensionalities: (1024, 1024, 1024) for the 2 hidden layers and output layer. We pre-train this network for 100 epochs.

For the comparisons in Sec. 3.5, we train a classifier stage (as defined in Table A.3), that operates on the output of the fixed network.

### A.1.7 Supervised AlexNet

For the results in this work, we tested two variations of the standard AlexNet architecture (one with Batch-Norm (Ioffe & Szegedy, 2015) and one without). Interestingly, although we found benefits of BatchNorm in convergence for our LCL training, we find that the standard AlexNet, without BatchNorm, provides a slightly better account of the neural data (and similar performance on the OOD behavioral benchmarks), As a result, use the standard network (without BatchNorm) for comparison. Because this network is fully-supervised on the ImageNet-1K recognition task (provided in the Pytorch library), we do not perform any extra training for the results in Sec. 3.5.

### A.1.8 L2-AT ($\epsilon = 3.0$)

We use an existing fully pre-trained version of the AlexNet architecture, trained with L2-AT adversarial training (Madry et al., 2017). We use the specific pre-trained model from Chen et al. (2020a), which uses $\epsilon = 3.0$ as the perturbation threshold. Because this network is fully-supervised on the ImageNet-1K recognition task (provided in the Pytorch library), we do not perform any extra training for the results in Sec. 3.5.

### A.1.9 ANT

We use the code provided at `https://github.com/bethgelab/game-of-noise` to train the standard AlexNet model with the adversarial noise training method provided in Rusak et al. (2020). This method differs from the L2-AT standard adversarial training as it uses a parameterized noise distribution (Gaussian) rather than arbitrary pixel perturbations within a fixed budget.

### A.2 Augmentation details

**Standard E2E training**

The standard E2E augmentation scheme that we use to both train the Barlow (E2E) (Zbontar et al., 2021) and Barlow (layerwise) (Siddiqui et al., 2023) baseline models is defined as follows: Each image is randomly augmented by composing the following operations, each applied with a given probability:

1. random cropping: a random patch of the image is selected, whose area is uniformly sampled in $[s \cdot \mathcal{A}, \mathcal{A}]$, where $\mathcal{A}$ is the area of the original image, and whose aspect ratio is logarithmically sampled in $[3/4, 4/3]$. $s$ is a scale hyper-parameter set to 0.08. The patch is then resized to 224 $\times$224 pixels using bicubic interpolation;

2. horizontal flipping;

3. color jittering: the brightness, contrast, saturation and hue are shifted by a uniformly distributed offset;

4. color dropping: the RGB image is replaced by its grey-scale values;

5. gaussian blurring with a 23$\times$23 square kernel and a standard deviation uniformly sampled from $[0.1, 2.0]$;

6. solarization: a point-wise color transformation $x \mapsto x \cdot \mathbb{1}_{x<0.5} + (1-x) \cdot \mathbb{1}_{x\geq0.5}$ with pixels $x$ in $[0, 1]$.

The augmented frames $x^A$ and $x^B$ result from augmentations sampled from distributions $\mathcal{A}_A$ and $\mathcal{A}_B$ respectively. These distributions apply the primitives described above with different probabilities, and different magnitudes. The following table specifies these parameters.

| Parameter | $\mathcal{A}_A$ | $\mathcal{A}_B$ |
|---|---|---|
| Random crop probability | 1.0 | |
| Flip probability | 0.5 | |
| Color jittering probability | 0.8 | |
| Color dropping probability | 0.2 | |
| Brightness adjustment max | 0.4 | |
| Contrast adjustment max | 0.4 | |
| Saturation adjustment max | 0.2 | |
| Hue adjustment max | 0.1 | |
| Gaussian blurring probability | 1.0 | 0.1 |
| Solarization probability | 0.0 | 0.2 |

**LCL-V2 Augmentations**

Given the limited capacity of our two-stage network, we drastically reduce the number and strength of the augmentation set. As described in the main text, we additionally complexity-matched the augmentations to generate the inputs used to train each layer. For our training, each image is first resized such that the smallest side is 224 pixels. A center crop is then selected from this image, and is randomly augmented by composing the following operations, each applied with a given probability.:

1. random cropping: a random patch of this central crop is selected, whose area is uniformly sampled in $[s \cdot \mathcal{A}, 0.9 \cdot \mathcal{A}]$, where $\mathcal{A}$ is the area of the original image, and whose aspect ratio is logarithmically sampled in $[0.9, 1.1]$. $s$ is a scale hyper-parameter set to a different value for each layer of the LCL-V2 architecture. The patch is then resized to $p \times p$ where $p$ is again dependent on the layer.

2. contrast and luminance jittering: the brightness and contrast are shifted by a uniformly distributed offset.

3. Gaussian noise: additive Gaussian noise is added independently to each channel of the RGB image. The noise is generated to be mean 0 with random standard deviation uniformly sampled from the range (0.04, 0.1).

The parameters to generate the first layer patches $x_1^A$ and $x_1^B$ are defined below:

| Parameter | $\mathcal{A}_A$ | $\mathcal{A}_B$ |
|---|---|---|
| Central crop size | 56 | |
| Minimum random crop scale | 0.6 | |
| Random crop resize output size | 48 | |
| Color jittering probability | 0.8 | |
| Color dropping probability | 0.2 | |
| Brightness adjustment max | 0.2 | |
| Contrast adjustment max | 0.2 | |
| Gaussian noise probability | 1.0 | 0.0 |

The parameters to generate the second layer patches $x_2^A$ and $x_2^B$ are defined below:

| Parameter | $\mathcal{A}_A$ | $\mathcal{A}_B$ |
|---|---|---|
| Central crop size | 112 | |
| Minimum random crop scale | 0.3 | |
| Random crop resize output size | 96 | |
| Color jittering probability | 0.8 | |
| Color dropping probability | 0.2 | |
| Brightness adjustment max | 0.2 | |
| Contrast adjustment max | 0.2 | |
| Gaussian noise probability | 1.0 | 0.0 |

## A.3 Neural evaluation details

### A.3.1 Stimulus pre-processing

We use the stimuli from Freeman et al. (2013); Ziemba et al. (2016), which consist of 225 naturalistic texture images from 15 texture families and 225 corresponding spectrally-matched noise images. In the experimental

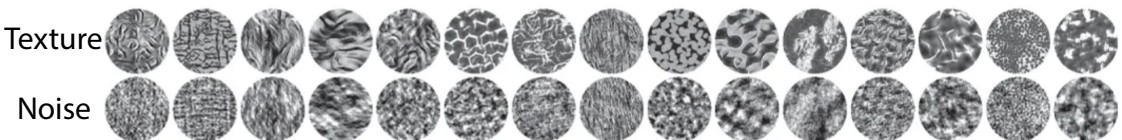

Figure A.1: Example samples of each of the 15 texture families and their corresponding spectrally-matched noise models. (Taken from Freeman et al. (2013)).

protocol used in Freeman et al. (2013), images were presented at a 4 deg field-of-view to the individual V1/V2 neurons. Since models have arbitrary input field-of-view (viewing distance), we use the pre-processing in the BrainScore benchmark `https://github.com/brain-score/brain-score`, which sweeps over field-of-view to find the optimal viewing distance for predicting the neural responses. This cross-validation is done on a held-out validation set. For example, for our model (and most ImageNet trained models), we find that an input size of 224 pixels and 8 degree field of view is optimal. As a result, the original texture stimuli are resized and placed in the central 4-degrees (112 pixel diameter). This image is then padded with gray values beyond the central 4 degrees.

### A.3.2 BrainScore neural prediction

Following the standard BrainScore pipeline, we preprocess the neural responses binning raw spike counts in 10ms windows. For each window over the 50ms to 150ms range post stimulus presentation, we average these spike counts over 20 repeated stimulus presentations. To evaluate neural predictivity we use the API provided in `https://github.com/brain-score/brain-score`. We show scores on the private split of the

data which consists of approximately 70% of the original images (official BrainScore split). Briefly, $N \times D_m$ model responses are regressed onto the $N \times D_n$ neural responses (V1: $D_n = 102$, V2: $D_n = 103$). This is done using the PLS regression method with 25 components. The regression is computed using 10-fold cross-validation and pearson correlations $r_n$ (between model predictions and neural responses across all test images) are obtained for each neuron and for each split. A measure of internal consistency $r_{ceil,n}$ is also computed by splitting neural responses in half across repeated presentations of the same image and computing the Pearson correlation coefficient between the two splits across images for each neuron. For more details, see Schrimpf et al. (2018).

For the overall scores presented in Fig. 4 (Left), The final explained variance for a given model is calculated as $median_n(r_n^2/r_{ceil,n}^2)$. For the subset explained variance scores in Fig. 4 (Right), these medians are computed over the specific subset of neurons identified as 'V1-like' or 'V2-like' (based on the V1-SteerPyr model explained variances).

However, medians over neurons can potentially obscure the performance of any given model so we also provide an additional figure comparing the per-neuron predictions of the LCL-V2 model against the L2-AT ($\epsilon = 3.0$) and Supervised AlexNet models.

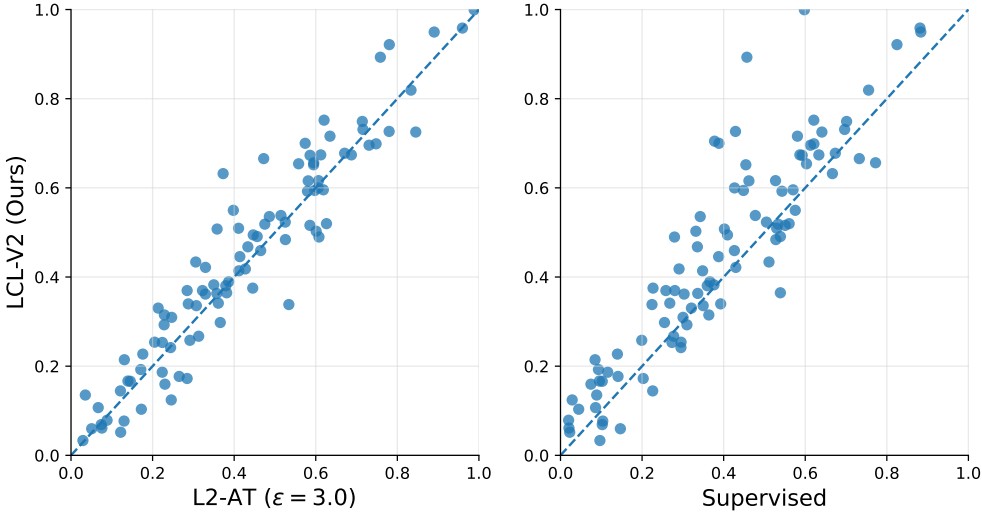

Figure A.2: **Comparing per neuron predictions via the BrainScore methodology.** We compare the explained variance (normalized by an estimated noise ceiling) *of each of the 103 V2 neurons* of our LCL-V2 model against other high-performing models. **Left:** LCL-V2 vs. L2-AT comparison shows that although there are a reasonable fraction of neurons which the L2-AT network predicts better ( 40%), there is a larger fraction ( 60%) which are better explained by our model. **Right:** LCL-V2 vs. Supervised comparison shows that for nearly all neurons (>75%), the explained variance of our model is comparable or greatly exceeds that of the Supervised network.

We further estimate the internal consistency between neural responses by splitting neural responses in half across repeated presentations of the same image and computing Spearman-Brown-corrected Pearson correlation coefficient between the two splits across images for each neuroid.

### A.3.3 Texture Modulation Analysis

The analyis on texture modulation uses the same stimuli and image pre-processing. For the neural responses, we follow the same pre-processing outlined above; however, instead of binning each neural responses over the fixed 50ms to 150ms window post presentation, we use the method in Freeman et al. (2013) which still selects a 100ms window (but now aligned to the specific response-latency of each neuron). This is done for consistency with the measured texture modulation indices in Freeman et al. (2013), but we do not find that it significantly changes the results.

As discussed above, for the regression analysis we provide explained variance scores relative to the noise ceiling (internal consistency). This is done to avoid biasing results based on predictions for neurons that are inherently unpredictable. However, for this analysis a noise ceiling on texture modulation cannot be calculated in the same way. For simplicity, we therefore calculate the texture modulations only for biological neurons which have an noise ceiling $R$ above a threshold (0.4). This excludes the bottom 10 (out of 103) neurons from the texture modulation calculation. Note, we find the results to be relatively robust to the choice of threshold.

Finally to test significance of the Spearman correlations between texture family modulation rank correlations, we use the following method outlined in Obilor & Amadi (2018):

We first transform the correlations to assume a normal distribution via Fisher's z-transformation Fisher et al. (1921): $z_i = arctanh(\rho_i)$. Given that our sample size is N=15 (15 texture families), we can calculate the standard error on the difference of correlations: $SE_{\delta_z} = \sqrt{\frac{2}{N-2}}$. Then, the test statistic $z_{test} = \frac{z_1 - z_2}{SE_{\delta_z}}$. Finally, the p-value can be calculated as $p = 2 * (1 - CDF(|z_{test}|))$.

### A.4 OOD and Human Behavior Benchmark

**Dataset information.** We evaluate accuracy and human error consistency using the code from `https://github.com/bethgelab/model-vs-human/tree/master`. We report the OOD accuracy (averaged over samples and distortions) on the full dataset from Geirhos et al. (2021), which consists of 17 OOD distortions applied to ImageNet images. These distortions include: cue-conflict, edge drawing, sketch drawing, silhouettes, texture stylization, three levels of eidolon noise, color jitter, false colors, contrast, low/high-pass noise, uniform noise, phase scrambling, spectral power equalization, and rotation. We additionally report the behavior error consistency metric on the recognition task with these distorted images.

**Behavior error consistency metric.** This metric, derived in Geirhos et al. (2020b) measures whether there is above-chance consistency with human per-sample recognition decisions. let $c_{h,m}(s)$ be 1 if both a human observer $h$ and $m$ decide either correctly or incorrectly on a given sample s, and 0 otherwise. The observed consistency $c_{h,m}$ is the average of $c_{h,m}(s)$ over all samples. The error consistency then measures whether this observed consistency is larger than the expected consistency given two independent binomial decision makers with matched accuracy (see Geirhos et al. (2020b) for the details on this exact computation).

### A.5 V1 Receptive Field Comparisons

In addition to the overall predictivity of the V1 data, we also use qualitative and quantitative analyses to probe the selectivities of the learned receptive fields in the first convolutional layer of our model. We compare these learned filters to those learned via adversarial-training and standard supervised training.

**Filter Visualization.** We visualize the 64 filters of each learned model in Fig. A.3. We visualize the filters in grayscale (even though the filters operate on color channels), to draw comparisons specifically between the spatial receptive field properties. By inspection, none of the models perfectly capture the nature of real V1 receptive fields; however, they all learn a set of reasonably diverse of multi-scale oriented filters. Our model and the L2-AT model seem to better capture the number of cycles within the oriented filters, whereas the supervised network receptive fields are tend to have high frequency and narrow bandwidth. Both LCL-V1 and Supervised models, however, seem to learn more localized non-oriented filters than the adversarially-trained model.

**V1 Receptive Field Properties.** We briefly explore some of the canonical receptive field properties highlighted in the data collected from macaque V1 neurons in Ringach (2002). For the three models, we fit a linear receptive field model to the filters shown in Fig. A.3 by minimizing the mean squared error between a parameterized 2-d Gabor function (product of a sinuoisoidal grating and a Gaussian envelope) and the model receptive field. We remove those cells that were not fit well.

We first measure the spatial phase of each receptive field and compare these to the distribution of spatial phase of the macacque data (Fig. A.4). None of the models perfectly match the neural data distribution, but our LCL-V1 model seems to best capture the relative bi-modal structure around even and odd-symmetry (0

LCL-V1 (Ours)          L2-Robust ($\varepsilon = 3$)          Supervised

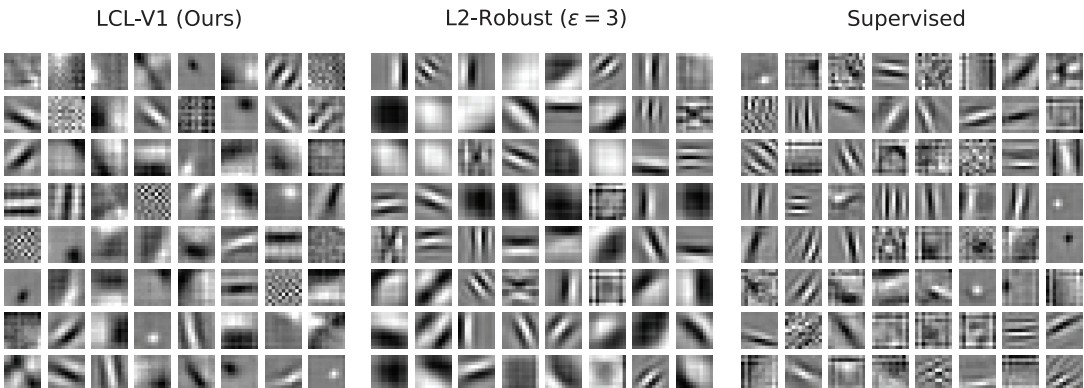

Figure A.3: **Comparing learned receptive fields.** The single-layer (layerwise) trained model (LCL-V1), learns a diverse set of oriented receptive fields. All 64 filters are shown for our method, adversarial robust training, and standard supervised training. All models learn a diverse set of filters, many of them oriented. Both our model and the adversarially-trained network learn more low-frequency filters than the standard supervised-trained network. Original filters are RGB, but we show grayscale versions here to focus on the spatial structure comparisons.

and $\pi/2$). We next compare the receptive field structure via the method in Ringach (2002), plotting the size of the receptive field envelope transverse to, and along its oscillating stripes ($(n_x, n_y)$, respectively), in units of the spatial period of the underlying sinusoid. We see in Fig. A.5, that while there are some outliers, our model receptive fields are distributed similarly to the V1 receptive fields measured in macaque. Compared with the other two models, our model also does a better job capturing the distribution of 'blob-like' filters near the origin.

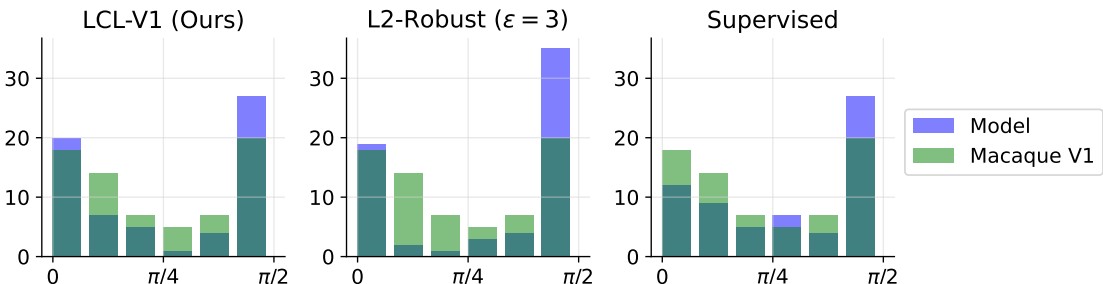

Figure A.4: **Comparing spatial phase selectivity between models and macaque V1 neurons.** We compare our LCL-V1 receptive field tuning to the equivalent receptive fields extracted from the L2-AT and Supervised networks. Model histograms of neuron spatial phase are shown in blue and real macaque data from Ringach (2002) is shown in green.

## A.6   Additional V2 BrainScore results

We provide additional more detailed results comparing the BrainScore explained variance of our LCL-V2 model against other top models on the BrainScore leaderboard. Many of these models use much higher capacity architectures, yet still underperform our model.

## A.7   Additional Human Behavior Results

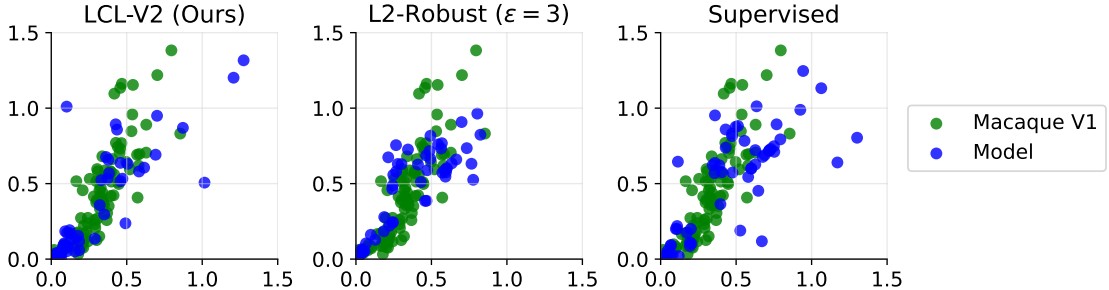

Figure A.5: **Comparison of learned oriented receptive field sizes to those measured physiologically.** Each point corresponds to a single cell (model, or biological), whose receptive field was fit with a Gabor function (product of a sinusoidal grating with a Gaussian envelope). Axes represent the standard deviation of the Gaussian envelope along the directions transverse to, and along, the sinusoidal stripes, in units of the sinusoidal period (referred to as $n_x$ and $n_y$, respectively, in Ringach (2002)). Model neurons are plotted in blue and Macaque V1 neurons in green.

| Method | V2 Ceiled Explained Variance ↑ |
|---|---|
| LCL-V2 | $\mathbf{0.417 \pm 0.008}$ |
| ResNet50 L2-AT ($\epsilon = 3.0$) (Madry et al., 2017) | $0.402 \pm 0.007$ |
| AlexNet L2-AT ($\epsilon = 3.0$) (Madry et al., 2017) | $0.397 \pm 0.007$ |
| VOneResNet-50 L2-AT (Dapello et al., 2020) | 0.380 |
| Bag-of-Tricks Model (Riedel, 2022) | 0.356 |
| AlexNet Supervised (Krizhevsky et al., 2012) | $0.353 \pm 0.009$ |
| ResNet-50 Supervised (He et al., 2016) | $0.320 \pm 0.01$ |
| ResNext-101 ($32 \times 8d$) WSL (Mahajan et al., 2018) | 0.312 |

Table A.4: **LCL-V2 outperforms all existing models for V2 data on BrainScore (including those that are non-architecturally matched).** The table compares performance in explaining macaque V2 responses for a variety of models that are either state-of-the-art BrainScore models (Bag-of-Tricks and ResNext-101 WSL) or prior state-of-the-art in V2 prediction (ResNet50 L2-AT). For those models for which we were able to obtain the full set of explained variance values, we provide standard deviation over 10 cross-validated splits.

| Method | IN-1K acc. ↑ | OOD acc. ↑ | obs. consistency ↑ | error consistency ↑ |
|---|---|---|---|---|
| LCL-V2Net | 0.527 | **0.492** | **0.643** | **0.211** |
| Barlow Twins (Zbontar et al., 2021) | 0.459 | 0.451 | 0.607 | 0.166 |
| Supervised (Krizhevsky et al., 2012) | **0.590** | 0.443 | 0.597 | 0.165 |
| VOneNet (Dapello et al., 2020) | 0.491 | 0.407 | 0.585 | 0.168 |
| L2-Robust (Madry et al., 2017) | 0.399 | 0.391 | 0.573 | 0.176 |

Table A.5: OOD accuracy and consistency with human judgments on 17 different OOD recognition tests, including multiple types of noise, phase-scrambling, and shape-biased stimuli. In both accuracy and human-alignment, our model trained with the LCL-V2 front-end improves over all end-to-end training approaches.

