# OpenReview forum: "Layerwise complexity-matched learning yields an improved model of cortical area V2"
_TMLR — Accepted by TMLR_

### Review · Reviewer_1T7Y · 2024-01-26

**Summary Of Contributions:**

The paper presents a novel approach to deep neural network training focused on layerwise complexity-matched learning (LCL). This methodology aligns the complexity of self-supervised learning objectives with the computational capacity of each DNN layer. The study showcases that LCL, when applied to a two-stage model, has better prediction of V2 neural responses. Additionally, when this model is used as a front-end for deeper networks in object recognition tasks, it exhibits improved out-of-distribution generalization and alignment with human behavior.

**Audience:**

Yes

**Claims And Evidence:**

Yes

**Requested Changes:**

Although the paper is well written overall, I would like to have the following points to be addressed:
1. In Figure 4, can you show the V2 Exp. Var. for all the 103 neurons (similar as figure 5)? I’d be interested in the distribution of all the 103 neurons rather than medians or median of medians.
2.  The description of Barlow twins objective function is confusing: according to the cited paper (Zbontar et al., 2021), b indexes batch sample (instead of batch), and the off-diagonal elements of the cross-correlation matrix “decorrelates the different vector components of the embedding... reduces the redundancy between output units” rather than “decorrelated features across the different images in a batch” or “decorrelating features across different images”

**Strengths And Weaknesses:**

Strengths:
* Innovative unsupervised approach to DNN training with layerwise complexity matching which demonstrates improved biological data alignment, out-of-distribution generalization and human behavior alignment.

Weaknesses:
* The study primarily focuses on a two-stage AlexNet model, it’s unclear whether it's effective for deeper and more advanced network architectures.
* Focused primarily on cortical area V2, with limited exploration of applicability to other areas.

---

> ### Author Response · Authors · 2024-04-25
> **Response to Reviewer 1T7Y**
>
> **Weaknesses: Study primarily focuses on a 2-stage architecture,  primarily focuses on V2**
>
> For the reasons noted in Section 2.4 we opted for a simpler network architecture that has the potential to be more interpretable. Additionally, in terms of neural predictivity simple architectures like AlexNet actually provide very competitive accounts of early visual areas when compared with architectures like ResNets and ViTs (this is evident in current BrainScore results). In order to provide a simple and fair comparison to alternatives, we locked the architecture for all of our analyses. We plan to explore extensions of this work to different architectures in future work, but note that the objective function and learning paradigm is general. Therefore, there are no reasons to believe that our approach will not scale well across architecture.
>
> Similarly, we wanted to focus the message of this work on capabilities that emerge in the second stage (the first is V1-like), so limited ourselves to 2 stages.  The objective is general, and should be applicable to later stages (something we will pursue in future work).
>
> V2, is a large visual area (as large as V1) but surprisingly remains poorly understood.  In particular, E2E recognition-trained deep nets can fit the data reasonably well, but have offered little insight into what the cells are computing.  On the other had, hand-constructed models have not been successful at fitting the data and provided little avenues to extending beyond very early vision (retina through V1).  As a result, we see our model as taking a notable step forward as it is simple and yet outperforms V2 predictions from much deeper networks. We hope that stacking on top of this model can provide a more fruitful path towards understanding of the computations being performed at multiple stages of the visual system.
>
> **Requested Changes**
>
> 1.  We have now included a plot of the per-neuron scores of our model (LCL-V2) vs the L2-AT and Supervised networks (See Fig. A.2). This plot shows that there is a wide distribution of scores across V2 neurons. What we can say right now is that compared with the L2-AT model, our model outperforms significantly on approximately 60% of neurons. The L2-AT model does out-perform ours on the other 40% but for the most part the two models are quite close on this subset. Compared to the Supervised network, our model outperforms much more significantly (approximately 75%) of neurons. It will be interesting to study this plot in more detail by analyzing what properties of the biological neurons indicate whether they will be well or poorly explained. We hope this might shed light on how to improve current models so we thank the reviewer for the suggestion.
>
> 2. We used the same exact form of the Barlow Twins loss as described in the original paper. As a result we have altered the description to be more clear and to align directly with that of the original paper. We hope this clarifies any confusion.

---

### Review · Reviewer_RvGn · 2024-02-13

**Summary Of Contributions:**

The paper describes a novel and effective way to train layers using only local (layer-specific) information rather than end-to-end backprop. This extra constraint on layer training yields a closer match to biological primate brain layers V1 and V2.

The paper also gives evidence that this training method, when it uses contrastive objectives, improves network performance in OOD generalization. This has high potential value in a pure ML sense (independent of any biological lens).

**Audience:**

Yes

**Claims And Evidence:**

Yes

**Requested Changes:**

As is usual (though lamentable) in reviews, to save time I have not commented on the many features of this work that I found admirable, clever, insightful, etc.

Minor comments:

pg 2: Spell out "IT"

section 2.3: typo "...augmented versions (x_2^A, x_2^B)..."

section 2.3: I did not see detail about the projection heads g_theta. Are these trainable, and what is their structure/architecture?

section "Neural alignment evaluations": This holdout set is qualitatively different from imageNet. How does this impact the results and/or their interpretation?

Fig 4, 6 (and other bar plots): can you include +/- std devs? This gives a quick sense of "significance" of any performance increases. For example, in fig 6, is the difference between LCL-V1 and BarlowTwins within a noise envelope?

Fig 4 caption: "of the best layer is provided". I was curious which layers turned out to be the best - were these rational/expected?

Fig 4 caption "The minimum explained .... 14%": These numbers don't seem to jibe with the right hand plot.

section 3.2, line 2: typos: "presentin", citations missing parentheses.

Fig 6: The "untrained" model seems to do remarkably well. Can you provide some detail as to what "untrained" means exactly, and comment on why it appears to do so well?

pg 12 last paragraph: typo "the the"

I loved the Discussion section - a good framing of the paper's context and implications.

Discussion, paragraph 2: Perhaps you can discuss how LCL relates to these other normative methods (sparsity etc), since these are all biologically-grounded. LCL also has biological grounding - are the methods mutually exclusive, or different lens on the same thing, or can they be combined, etc.

Limitations etc: "These networks contain additional computational elements ...". Are structures like skip connections supported by biology findings, or is this on a separate track for future work, viz pure ML-centric development as opposed to seeking alignment with biology. Both tracks are valid directions given the findings in the paper.

**Strengths And Weaknesses:**

Reviewer limitations:

1. I apologize to the authors and AE for my lateness turning in a review.
2. I am not versed in this slice of literature, so my estimate of the paper's grounding in the literature is based on its internal evidence.


Comment to TMLR:

Line numbers make review much easier. Can these be added to the template?


General comments:

This is a very high quality research program and paper, the kind of thing that I wish I was grown-up enough to produce myself.
It is well written and easy to follow.
It effectively connects and combines neuroscience and ML perspectives.
It effectively describes and builds on existing literature.
It offers novel and valuable advances, relevant to both biological and ML-centric perspectives.

Is there a publicly-available codebase? That is always a good feature.

---

> ### Author Response · Authors · 2024-04-25
> **Response to Reviewer RvGn Requested Changes**
>
> Thank you for the notes on typos- we have fixed these in the revised manuscript. We do intend to release a codebase and a way to reproduce the results on publication of the paper.
>
> **section 2.3: I did not see detail about the projection heads g_theta.**
>
> The exact implementation of the projection heads are described in Appendix A.1.1. They are an MLP with 1 hidden layer and are trainable. This is very similar to the projection heads used in Barlow Twins, SimCLR, BYOL etc.
>
> **Neural alignment holdout set being different from ImageNet**
>
> This is a very good and interesting point. Unfortunately, most neural datasets do not cover the same kind of distribution or diversity as ImageNet. Overall, I think this is good because the neural evaluation is in effect testing how well your model can capture OOD responses of neurons. Because ImageNet is far more diverse than the set of textures used in the evaluation, a model trained on ImageNet should be able to generalize to this more restricted subset of textures. Models that do not, most likely are overfitting to the stimuli they are trained on and thus are not good models of the underlying biology.
>
> That being said, we would like to clarify the setup in case there was any confusion. Models are all trained on ImageNet images, but during the neural alignment evaluation, we train a linear regression on a subset of the texture images and test the regression on a held-out set. Therefore, the regression is learned on a distribution similar to the final test evaluation.
>
> **Bar Plot std-devs**
>
> Thank you for this note- we have added standard deviation ticks to the bar plots to show significance. Overall, the standard deviations across cross-validated splits (10 folds) are quite small and so our results and interpretation does not change much.
>
> **Fig 4 caption: Chosen layers for BrainScore**
>
> Obviously for our model we fixed the layers to the two layers in our model (first layer predicts V1 and second layer predicts V2). In the pre-trained E2E networks we did look into which layers get chosen and these results are a bit more mixed. For some training objectives similar layers (first and second) get chosen but for other objectives it seems that deeper layers provided better accounts of the data. We think this is potentially a problem with E2E objectives because there are no constraints on intermediate layers. As a result, we hypothesize that a full layerwise method like ours would result in more reasonable layer-selections.
>
> **Fig 4 caption: minimum explained variance**
>
> Thank you, we have fixed this- the number was 4%
>
> **Fig 6 untrained model performance**
>
> The untrained model does indeed perform quite well in the V1 prediction task. We believe this has to do with the fact that we are using a neural alignment method that relies on linear regression. Because deep networks have thousands of neurons in a given layer, they are quite high-dimensional. Therefore, given enough random high-dimensional bases, it is possible to linear combine these bases to predict any response spanned by these bases. Since V1 neurons often encode very simple transformations on the input image, a random linear combination of neurons can in fact perform reasonably well. This becomes less true as you try to predict responses of neurons in deeper layers such as V2, V4, IT. In summary, this result is really an artifact of the regression methodology and so in future work it would be worth using a more diverse set of metrics to compare models to brain responses.
>
> **Discussion comments on relation to sparsity etc.**
>
> Thank you for the positive view on our Discussion. We have tried to add additional commentary on the relationship between our method and prior normative principles like sparsity etc. The core of our method relies on the layerwise complexity matching idea and leveraging redundancy reduction in the context of the Barlow Twins loss. However, complexity-matching is not inherently tied to this specific normative objective function so it is very well possible that there are ways to create objectives driven by sparsity, temporal prediction, etc that also leverage a layerwise complexity-matching framework to improve their performance.
>
> **Limitations**
> Skip connections are indeed supported by biological findings. For example V1 has direct projections to both V2 and  V4, so future work including these would indeed be valuable to both ML and application to biological alignment. In V2 specifically, most of the feed-forward projections come straight from V1 so there was less need to model skip connections. However, in extending our model to V4, IT etc. we would definitely need to consider this carefully. That being said, as you noted, the inclusion of skip connections and deeper networks would also just have direct ML applications as well.

---

### Review · Reviewer_SmQ8 · 2024-04-03

**Summary Of Contributions:**

The paper presents a layer-wise self-supervised learning paradigm that results in internal representations which are better aligned with visual area V2 than alternative supervised, self-supervised and layer-wise training paradigms using the AlexNet architecture. When using two layers trained in this way as a fixed frontend to AlexNet, it improves out-of-domain generalization and alignment of model predictions with humans in image classification.

**Audience:**

Yes

**Broader Impact Concerns:**

None.

**Claims And Evidence:**

Yes

**Requested Changes:**

Overall the paper is well organized, well written, easy and fun to read, and makes a very nice contribution. I really enjoyed reading the paper. Nevertheless, I have a few concerns, which I would like to detail below.


### 1. Is TMLR the right venue?

The paper has great potential for moving vision science and computational neuroscience forward. It could also impact machine learning as a field, but in this regard the contribution is less clear. I therefore wonder whether TMLR is the right venue for this paper. While I leave a specific recommendation/decision to the editorial board and/or the authors, let me explain my reasoning:

 - I see the main contribution of the paper as providing a potential normative account of primate area V2, which is also evident in most of the results figures (4–7) being related to model-neuron alignment.

 - The improved OOD robustness and human alignment (Fig. 8) is of potential interest to the ML community, but it feels more like a corollary than the main result and is only a teaser. The main questions from an ML perspective would be whether this approach (a) can be scaled to more than two layers and (b) also holds for more modern encoder architectures. While the authors acknowledge these limitations, the defer them to future work.



### 2. Some methodological details are unclear

There are a few methodological details that I consider somewhat important, but could not resolve from reading the text:

 1. Do I understand the equation on p. 5 correctly that "Pool" refers to a global average (or max?) pooling, which contracts the spatial dimensions to $1 \times 1$? As the input (for layer 1) is $48 \times 48$ and the conv kernels are $ 11 \times 11$ with stride 4, it would reduce to $9 \times 9$ after the convolution, then to $4 \times 4$ after maxpool, which then gets reduced to $1 \times 1$ by the global average pooling? In other words, does $f_{\theta_1}$ include the $3 \times 3$ max pooling with stride 2 of AlexNet's first layer? Is the same true for the second layer, $f_{\theta_2}$? I assume it must, because otherwise it could not be used as a frontend to the later layers of AlexNet for Fig. 8? It would be good to be more explicit on this in the manuscript.

 1. The training as depicted in Fig. 3 suggests that both layers are being trained jointly and only the gradient is not passed backwards from layer 2 to layer 1. Is this important or could you also train sequentially where you first optimize the parameters of layer 1 until convergence and then start with layer 2?

 1. How did you choose the input resolution? Cortical receptive fields scale with the eccentricity of the neuron's receptive field, whereas a convnets' receptive field size is given by the kernel size and the learned features depend on the image size and resolution. As e.g. shown by Cadena et al, PLoS CB 2019, the input resolution has a pretty substantial effect on the neural predictivity. How do your results and conclusions change when you increase or decrease the image resolution by, e.g., 50%?

 1. Related to the previous point, I would like to ask whether the (effective) image resolution (after data augmentation) was the same for layer 1/V1 and layer 2/V2 training. I could not completely understand the data augmentation pipeline and how exactly the different scaling/cropping parameters are used for layer 1 vs. layer 2. Suppose you input a single image with an object of, say, size 20 pixels repeatedly to the data augmentation pipeline. Will the distribution of resulting object sizes after applying the data augmentation be identical for layer 1 and layer 2 training? If not, wouldn't this affect your conclusions? Also, if not, how would this be implemented in the brain, where there is only one set of inputs to the eye?

 1. How exactly did you compute the texture modulation index presented in Fig. 5? The way it is defined in the methods section, it should be between –1 and 1, but the figure shows values roughly between –4 and 4. I can see two ways how this could happen: (a) it was normalized/standardized prior to analysis, in which case it would be good to state what exactly was done; (b) activation values could be negative, in which case the denominator could get arbitrarily small and close to zero, which would mean the index is not well defined (it makes sense only for strictly non-negative quantities) and the whole analysis doesn't really make sense.



 ### 3. Comparison to alternative accounts of V2

Could your model be compared to Freeman & Simoncelli's texture model of V2? You present clear evidence that your approach outperforms your baselines, but they are not particularly strong baselines for V2. The paper would be more convincing if it presented some evidence that your account of V2 is competitive with or even better than Freeman & Simoncelli's model, which they showed can account for the texture selectivity of V2 neurons.



 ### Minor comments:

 - The first reference to Fig. 3 at the beginning of Section 2 seems too early. If the figure is relevant there already, why not move it there and call it Fig. 2? If it's not relevant there, why cite it and make readers search for it?

 - Fig. 2 bottom: Strictly speaking only $f_{\theta_l}$ depends on $L_l$, not $f_{\theta_{1,...,l}}$, correct?

 - P.6 "Additionally, as mentioned earlier, much of the biological anatomy and computational theories suggest that the feed-forward aspect of areas V1 and V2 should be explainable by networks with few computational stages. As a result, we hypothesize that the AlexNet architecture can provide a more parsimonious and interpretable model of these areas." Could you provide references for this claim? Cadena et al. 2019 [1] suggest it's quite a few nonlinearities already for V1, and Miao & Tong 2023 [2] show that AlexNet needs to be at least modified quite a bit to get there. Also, your explained variance for V1 is only around 50% of the top scores on BrainScore, arguing somewhat against your claim. Maybe I'm overlooking additional evidence that backs your claim or I misunderstood it?

 - Fig. 4+5+8: What does the asterisk signal? It's usually used to depict statistical significance, but I cannot find any information on that.

 - Section 3.2, p. 10: Can you provide statistics on the difference of rank correlations between the three models? With n=15, it appears to me quite plausible that a difference of 0.59 vs. 0.8 might arise by random chance.

 - Fig. 8: If using a bar plot, please start the y axis at zero. Otherwise it's hugely misleading.


 ### Requested changes

I would like the authors to address my main concerns about lack of clarity regarding the methodology (point 2). Regarding point 3, it would be great if they could provide a comparison, but I realize that it may be challenging and therefore would not insist on such a comparison.



 ### References


[1] Cadena et al., PLoS CB 2019, https://journals.plos.org/ploscompbiol/article?id=10.1371/journal.pcbi.1006897

[2] Miao & Tong, bioRxiv 2023, https://www.biorxiv.org/content/10.1101/2023.08.26.554952v1

**Strengths And Weaknesses:**

### Strengths

 1. Paper is well written and easy to read
 1. Presents convincing evidence for its claims
 1. Repeated stacking of normative approaches such as redundancy reduction or sparse coding has thus far not been successful, making the success of this approach a clear strength



### Weaknesses

 1. Not sure if TMLR is the right venue for this work
 1. Some methodological details are unclear from the text and could undermine the claims
 1. No comparison to or discussion of relationship to V2 model by Freeman & Simoncelli

---

> ### Author Response · Authors · 2024-04-25
> **Response to Reviewer SmQ8: Is TMLR the right venue?**
>
> As you mention as a strength of our work, layerwise learning of cascades has not (to date) worked well for sparse coding, redundancy reduction (or any other biologically-inspired objective that we're aware of).  In particular these methods have not been able to scale beyond small low-resolution image patches or datasets (such as cifar, mnist etc.). As a result, the fact that our method is able to overcome these limitations (albeit in just two stages) we believe is a significant advance for the layerwise/local learning community. In particular, we think our framework around ‘complexity-matching’ can be relevant to the broad machine learning community even though our primary "application" is to neuroscience. Furthermore, as you say, we have explored preliminary directions on how layerwise learned models can impact ML directly (through our OOD robustness and human-alignment studies). While these might not be the main results, we also don’t think they should be overlooked as improvements on the Geirhos dataset benchmarks are hard to achieve even with more modern architectures. In particular, we achieve better human-error-consistency (21.1) using our LCL-V2-AlexNet model vs. a supervised ResNet-50 mode (20.8) or supervised ViT-L (20.6)).

---

> > ### Author Response · Authors · 2024-04-25
> > **Response to Reviewer SmQ8: Methodological details**
> >
> > **Confusion over Pooling in p.5**
> >
> > We have revised this explanation in the main text as requested. To be clear, the pooling is global average pooling (GAP) which we apply to the projected features (at each spatial location) to obtain a single projection embedding per example. Because this is applied after the projection head, it is only used during training. During evaluation or when using the model as a front-end for the OOD experiments, we use the AlexNet architecture (with MaxPooling as you described) in the first two stages. We have also provided details on the architecture and projection heads in the appendix.
> >
> > **Joint vs greedy training**
> >
> > This is a very interesting question and something we do want to explore in future work. Intuitively, we believe the result would not change much (or may even improve) if we trained the first stage to convergence before training the second stage. From the perspective of biological plausibility, we feel the joint training is more plausible as it doesn't require neurons in downstream layers to know about convergence of prior layers.
> >
> > **Input resolution questions**
> >
> > Input resolution, as you suggest is largely a free-parameter in the methodology for comparing CNNs to biological neurons. More specifically, “viewing distance” of a model cannot be determined which means that the pixels/deg field-of-view ratio is arbitrary for any model. However, for biological neurons, the datasets are collected with very specific knowledge of the field-of-view. For example, in our evaluation we use texture images that were shown to V1/V2 neurons such that the images covered a 4-degree field of view. Therefore, assuming a given field-of-view for a model, images must be presented such that they match the presentation to the biological neurons. For example if a model uses 224x224 pixel images and you assume an 8-deg field of view, then you would have to resize the texture images and place them in a central 112x112 pixel aperture with gray pixel padding. On the other hand if you assume the model sees a 4-deg field of view, then you would resize the texture image to the full 224x224 pixel input. As described in the Cadena paper, for any given model, this resizing will change the best prediction layer (as the effective receptive field sizes will change) and potentially the best score. Following the BrainScore paradigm, we perform a sweep over field-of-view: (2deg, 4deg, 6deg, 8deg, 10deg, 12deg etc) and pick the one that performs best (on a held-out validation set). We then use this setting for the final evaluation on the private test set. We see this as putting all models on an even playing field since we are taking the best setting of resolution / field-of-view for each model and not imposing any common resolution across model evaluations.
> >
> > Regarding the training of our model, you make a good point. To describe the procedure again, consider an input that covers an object at (224 pixels). The first layer takes as input a crop of 56 pixels (reducing the scale of content from an object to a small part of an object). This cropping controls the complexity of the content but does not change the effective resolution (i.e. the same content covers the same amount of pixels, we are just restricting the scale of content seen by a given layer). A similar procedure is applied for the second layer but now with a central 112x112 crop. Again this larger crop covers a larger part of the original input (i.e. higher complexity of content), but does not change effective image resolution. We then apply light spatial and photometric distortions to the image patches for each layer to get the second view, but critically we scale these augmentations such that again the effective relative image resolution between layer 1 and layer 2 does not change. So to be clear, an object that covers a certain set of pixels in the input will still cover the same pixels in the input to layer 1 and layer 2. However, what is changing is the total span of the content seen by layer 1 vs layer 2. This change in content complexity is a main driver of the result as it determines what invariances get learned by our model.
> >
> > That being said, the biological plausibility of this crop and augmentation scheme is something that we intend to fix in future work. Specifically, we believe there is an implementation analogous to our method that uses a single input image to both layers but does the extraction of spatial neighborhoods (crops) in feature space in order to do obtain the “augmented views”.
> >
> > **Computation of texture modulation index presented in Fig. 5?**
> >
> > Thank you very much for pointing this out. There in fact was a bug in the code used to generate the texture modulations (as it was doing standardization implicitly). We have now fixed this an updated figure 5 (please see the updated paper for this).

---

> > > ### Author Response · Authors · 2024-04-25
> > > **Response to Reviewer SmQ8: Alternative accounts of V2 (Freeman Simoncelli)**
> > >
> > > The Freeman & Simoncelli work (and the Portilla & Simoncelli texture model on which it was based) are based on a set of hand-constructed higher-order statistics.  These were used to generate images (samples with matching statistics), which demonstrate that the statistics capture a substantial amount of structure in natural textures, and are relevant for human vision. However these statistics do not provide an explicit model for neural representation in V2 and there were no explicit predictions of V2 responses in this work.
> > >
> > > The combination of the Freeman 2013 work and Ziemba 2017 work that we cite (on neural representation of V2) does indeed show that V2 neurons do exhibit selectivity for natural textures; however, they also do find a substantial number of neurons that are not well modulated by texture. Therefore, we hypothesize that a model solely focused on representing texture would miss critical aspects of V2 selectivity. In terms of image-computable, learned models of V2, our best set of comparisons are those submitted to the BrainScore platform and among these our model is the top performing model.
> > >
> > > That being said, we do appreciate the note and do plan to see if our model captures similar statistics to the the hand-crafted Portilla texture model.

---

> > > > ### Author Response · Authors · 2024-04-25
> > > > **Response to Reviewer SmQ8: Minor comments**
> > > >
> > > > **The first reference to Fig. 3 at the beginning of Section 2 seems too early.**
> > > >
> > > > Thank you. we have fixed this.
> > > >
> > > > **Fig. 2 bottom**
> > > > Yes strictly speaking you are correct that the parameters ${f_\theta}_l$ only depend on $L_l$. We use that diagram simply to demonstrate that the loss at that layer $l$ must be matched in complexity to the model capacity covering all layers up to $l$ because the representation is still being computed as a cascade. It is just that the parameters of stage $l$ are being updated with respect to the per-layer loss.
> > > >
> > > > **P.6 AlexNet and simple models of V1/V2**
> > > >
> > > > We are sorry for the confusion and have tried to update the text in our revised manuscript to be more clear. It is not that we do not believe there are extra non-linearities involved in truly capturing V1 responses. It is well-known that divisive normalization, recurrence, etc all matter. However, we also do know based on the current BrainScore results and the VOneNet work (Dapello et al. 2020) that a feed-forward model with filters that tile orientation and spatial frequency space effectively can be a very strong baseline model for V1. In fact even the Miao & Tong 2023 work that you refer to does show that compared with a deeper VGG network, shallower networks already does surprisingly well if the filter sizes etc are adjusted accordingly. While ideally, we would have liked to start with a more biologically-plausible architecture that best fit V1, our goal was to compare our learning objective to end-to-end trained networks without having to re-train many of them ourselves. Therefore, AlexNet provided a reasonable baseline with many pre-trained public checkpoints that we could compare to.  It is also worth noting that on BrainScore, while it is far from the best model, AlexNet does better as a baseline for V1/V2 prediction than much larger ResNets or ViTs so we decided that the trade-off of performance vs simplicity was worth it as a starting point.
> > > >
> > > > Regarding our V1 results: the 50% drop is not accurate as the BrainScore numbers are a combination of V1 PLS regression predictions and a collection of single neuron V1 tuning property benchmarks. For simplicity we primarily focused on the regression benchmarks and on these, our V1 model is near the top of "architecture-matched" models and captures approximately 80% of the variance of the top performing hand-crafted models (which we show in Figure. 6)
> > > >
> > > > An additional reason for focusing on a simple two-stage model without many more non-linearities is that we know that V2 primarily takes feed-forward projections directly from V1 [1]. Therefore, it is hypothesized that a large fraction of V2 response properties should be able to be explained by a model that performs a simple computation given V1-afferents. Having arrived at a V2 model that is in fact performing a single-stage of computation on top of a V1-like front-end, we do believe we might be able to probe the model to better understand the underlying computations of V2 (something that would be far more difficult in an arbitrarily deep network).
> > > >
> > > > **Fig. 4+5+8: What does the asterisk signal?**
> > > >
> > > > We had initially been using this to denote the best model, but realize it is confusing and have removed these asterisks. Thank you!
> > > >
> > > > **Section 3.2, p. 10: Statistics on the rank correlation**
> > > >
> > > > We have now updated the text with a statistical analysis on the rank correlations. Specifically we computed the rank correlations with the neural data family ranks using 50000 randomly sampled orderings of the 15 ranks. We used this sample to compute 95% confidence intervals on the expected correlation by chance. This confidence interval is [0.54, -0.54]. The two models we compare to (L2-AT and Supervised) are actually either within this interval or just slightly above whereas our correlation of 0.84 is significantly higher (even higher than a 99% confidence interval). This indicates that our model is the only one that is producing a rank correlation signficantly higher than that expected by chance.
> > > >
> > > >
> > > > Fig. 8: If using a bar plot, please start the y axis at zero. Otherwise it's hugely misleading.
> > > >
> > > > Thank you for the note. We never meant to be misleading but just wanted the reader to be able to see the differences clearly. However we understand that this can be misleading and have altered the bar plot to start at 0 now.
> > > >
> > > > [1] Yasmine El-Shamayleh, Romesh D Kumbhani, Neel T Dhruv, and J Anthony Movshon. Visual response
> > > > properties of v1 neurons projecting to v2 in macaque. Journal of Neuroscience, 33(42):16594–16605, 2013

---

> > > > > ### Comment · Reviewer_SmQ8 · 2024-04-26
> > > > > **Rank correlations**
> > > > >
> > > > > Just a small follow-up: I am not sure I understand correctly how you assessed whether the difference between a rank correlation of 0.59 and 0.84 is statistically significant. It seems you tested individually for each sample whether its correlation is greater than expected by chance and found that this is only the case for one of the methods. Unfortunately, from this analysis you cannot deduce that two measured correlations are actually different. That's a classical fallacy in statistical testing.
> > > > >
> > > > > In fact, when I interviewed ChatGPT citing your numbers, it suggests the p value for the difference is larger than 0.05: https://chat.openai.com/share/9b571e88-b630-45a9-9b4d-9901d203135c
> > > > >
> > > > > (Disclaimer: I did not verify its calculations in detail, but would have expected your argument to follow the lines of ChatGPT's answer)

---

> > > > > > ### Author Response · Authors · 2024-05-09
> > > > > > **Response regarding rank correlation statistical significance**
> > > > > >
> > > > > > Hi thank you very much for this comment very sorry for the delayed response (for some reason the notification email was missed regarding this post).
> > > > > >
> > > > > > We mis-read your initial comment and accordingly have now added the statistical significance test based on the Fisher z-transformation that is essentially following the methodology you outlined in your post.
> > > > > >
> > > > > > Note, in going back through this analysis, we also realized that there are some highly noisy biological neurons which we were computing texture modulation indices for (thus biasing the ground truth data). In the regression analyses, this was accounted for by ceiling the predictions by the noise-ceiling (R^2) for each neuron based on internal consistency (predicting neural firing using held-out trials for the same image). However, there is no easy way to correct the texture modulation measures by a noise ceiling, so we used a threshold to exclude highly noisy biological neurons (~10 out of the 103 V2 neurons) from the calculation of the ground-truth texture modulations. Once we do this, we in fact see that our model's predictions get even better and the new correlations are (0.9 vs 0.56 and 0.58). At these correlations, both comparison are in fact statistically significant even given 15 samples.
> > > > > >
> > > > > > We hope this addresses the reviewer's concerns.

---

> ### Comment · Reviewer_RvGn · 2024-04-25
> **props**
>
> re SmQ8: Thank you for your wonderful, thoughtful review.

---

### Author Response · Authors · 2024-04-25
**Updated revision and response for all reviewers**

We thank all reviewers for their generally positive comments about both the novelty and presentation of our work. We have updated our paper and uploaded a revision that addresses many of the comments and critiques mentioned by all reviewers. We hope that this demonstrates our intention to take all comments seriously into consideration to improve the final submission. We refer each reviewer to our specific responses below and are happy to make any further changes requested.

---

### Decision · Action_Editor_nexS · 2024-05-14

**Recommendation:** Accept as is

**Comment:**

All reviewers agree that the paper is suitable for publication in TMLR.
I propose a featured certification for the submission. The paper is very well-written with creative and solid technical content. It includes a conceptual advance for layer-wise self-supervised learning that I believe can be highly interesting for the TMLR audience interested in local learning with brain-inspired algorithms.  The proposed model improves out-of-domain generalization and model predictions are well-aligned with data from humans in image classification.

**Audience:**

The reviewers agree that the paper is of interest in particular for researchers interested in local learning with brain-inspired algorithms.

**Claims And Evidence:**

The claims made by the submission are supported by clear convincing evidence.